# Model Reconstruction Using Counterfactual Explanations: A Perspective From Polytope Theory

**Pasan Dissanayake**
University of Maryland
College Park, MD
pasand@umd.edu

**Sanghamitra Dutta**
University of Maryland
College Park, MD
sanghamd@umd.edu

## Abstract

Counterfactual explanations provide ways of achieving a favorable model outcome with minimum input perturbation. However, counterfactual explanations can also be leveraged to reconstruct the model by strategically training a surrogate model to give similar predictions as the original (target) model. In this work, we analyze how model reconstruction using counterfactuals can be improved by further leveraging the fact that the counterfactuals also lie quite close to the decision boundary. Our main contribution is to derive novel theoretical relationships between the error in model reconstruction and the number of counterfactual queries required using polytope theory. Our theoretical analysis leads us to propose a strategy for model reconstruction that we call Counterfactual Clamping Attack (CCA) which trains a surrogate model using a unique loss function that treats counterfactuals differently than ordinary instances. Our approach also alleviates the related problem of *decision boundary shift* that arises in existing model reconstruction approaches when counterfactuals are treated as ordinary instances. Experimental results demonstrate that our strategy improves fidelity between the target and surrogate model predictions on several datasets.

## 1 Introduction

Counterfactual explanations (also called *counterfactuals*) have emerged as a burgeoning area of research [Wachter et al., 2017, Guidotti, 2022, Barocas et al., 2020, Verma et al., 2022, Karimi et al., 2022] for providing guidance on how to obtain a more favorable outcome from a machine learning model, e.g., increase your income by 10K to qualify for the loan. Interestingly, counterfactuals can also reveal information about the underlying model, posing a nuanced interplay between model privacy and explainability [Aïvodji et al., 2020, Wang et al., 2022]. Our work provides the first theoretical analysis between the error in model reconstruction using counterfactuals and the number of counterfactuals queried for, through the lens of polytope theory.

Model reconstruction using counterfactuals can have serious implications in Machine Learning as a Service (MLaaS) platforms that allow users to query a model for a specified cost [Gong et al., 2020]. An adversary may be able to "steal" the model by querying for counterfactuals and training a surrogate model to provide similar predictions as the target model, a practice also referred to as *model extraction*. On the other hand, model reconstruction could also be beneficial for *preserving applicant privacy*, e.g., if an applicant wishes to evaluate their chances of acceptance from crowdsourced information before formally sharing their own application information with an institution, often due to resource constraints or having a limited number of attempts to apply (e.g., applying for credit cards reduces the credit score [Capital One, 2024]). Our goal is to formalize *how faithfully can one reconstruct an underlying model given a set of counterfactual queries.*

38th Conference on Neural Information Processing Systems (NeurIPS 2024).

An existing approach for model reconstruction is to treat counterfactuals as ordinary labeled points and use them for training a surrogate model [Aïvodji et al., 2020]. While this may work for a well-balanced counterfactual queries from the two classes lying roughly equidistant to the decision boundary, it is not the same for unbalanced datasets. The surrogate decision boundary might not always overlap with that of the target model (see Fig. 1), a problem also referred to as *a decision boundary shift*. This is due to the learning process where the boundary is kept far from the training examples (margin) for better generalization [Shokri et al., 2021].

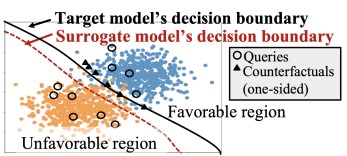

Figure 1: Decision boundary shift when counterfactuals are treated as ordinary labeled points.

The decision boundary shift is aggravated when the system provides only *one-sided counterfactuals*, i.e., counterfactuals only for queries with unfavorable predictions. If one were allowed to query for two-sided counterfactuals, the decision boundary shift may be tackled by querying for the counterfactual of the counterfactual [Wang et al., 2022]. However, such strategies cannot be applied when only one-sided counterfactuals are available, which is more common and also a more challenging use case for model reconstruction, e.g., counterfactuals are only available for the rejected applicants to get accepted for a loan but not the other way.

In this work, we analyze how model reconstruction using counterfactuals can be improved by leveraging the fact that the counterfactuals are quite close to the decision boundary. We provide novel theoretical analysis for model reconstruction using polytope theory, addressing an important knowledge gap in the existing literature. We demonstrate reconstruction strategies that alleviate the decision-boundary-shift issue for one-sided counterfactuals. In contrast to existing strategies Aïvodji et al. [2020] and Wang et al. [2022] which require the system to provide counterfactuals for queries from both sides of the decision boundary, we are able to reconstruct using only one-sided counterfactuals, a problem that we also demonstrate to be theoretically more challenging than the two-sided case (see Corollary 3.8). In summary, our contributions can be listed as follows:

**Fundamental guarantees on model reconstruction using counterfactuals:** We derive novel theoretical relationships between the error in model reconstruction and the number of counterfactual queries (query complexity) under three settings: (i) Convex decision boundaries and closest counterfactuals (Theorem 3.2): We rely on convex polytope approximations to derive an *exact* relationship between expected model reconstruction error and query complexity; (ii) ReLU networks and closest counterfactuals (Theorem 3.6): Relaxing the convexity assumption, we provide *probabilistic* guarantees on the success of model reconstruction as a function of number of counterfactual queries; and (iii) Beyond closest counterfactuals: We provide approximate guarantees for a broader class of models, including ReLU networks and locally-Lipschitz continuous models (Theorem 3.10).

**Model reconstruction strategy with unique loss function:** We devise a reconstruction strategy – that we call Counterfactual Clamping Attack (CCA) – that exploits only the fact that the counterfactuals *lie reasonably close to the decision boundary, but need not be exactly the closest.*

**Empirical validation:** We conduct experiments on both synthetic datasets, as well as, four real-world datasets, namely, Adult Income [Becker and Kohavi, 1996], COMPAS [Angwin et al., 2016], DCCC [Yeh, 2016], and HELOC [FICO, 2018]. Our strategy outperforms the existing baseline [Aïvodji et al., 2020] over all these datasets (Section 4) using one-sided counterfactuals, i.e., counterfactuals only for queries from the unfavorable side of the decision boundary. We also include additional experiments to observe the effects of model architecture, Lipschitzness, and other types of counterfactual generation methods as well as ablation studies with other loss functions. A python-based implementation is available at: `https://github.com/pasandissanayake/model-reconstruction-using-counterfactuals`.

*Related Works:* A plethora of counterfactual-generating mechanisms has been suggested in existing literature [Guidotti, 2022, Barocas et al., 2020, Verma et al., 2022, Karimi et al., 2022, 2020, Mothilal et al., 2020, Dhurandhar et al., 2018, Deutch and Frost, 2019, Mishra et al., 2021]. Related works that focus on leaking information about the dataset from counterfactual explanations include membership inference attacks [Pawelczyk et al., 2023] and explanation-linkage attacks [Goethals et al., 2023]. Shokri et al. [2021] examine membership inference from other types of explanations, e.g., feature-based. Model reconstruction (without counterfactuals) has been the topic of a wide array of studies (see surveys Gong et al. [2020] and Oliynyk et al. [2023]). Various mechanisms such as equation solving [Tramèr et al., 2016] and active learning have been considered [Pal et al., 2020].

Model inversion [Gong et al., 2021, Struppek et al., 2022, Zhao et al., 2021] is another form of extracting information about a black box model, under limited access to the model aspects. In contrast to model extraction where the goal is to replicate the model itself, in model inversion an adversary tries to extract the representative attributes of a certain class with respect to the target model. In this regard, Zhao et al. [2021] focuses on exploiting explanations for image classifiers such as saliency maps to improve model inversion attacks. Struppek et al. [2022] proposes various methods based on Generative Adversarial Networks to make model inversion attacks robust (for instance, to distributional shifts) in the domain of image classification.

Milli et al. [2019] looks into model reconstruction using other types of explanations, e.g., gradients. Yadav et al. [2023] explore algorithmic auditing using counterfactual explanations, focusing on linear classifiers and decision trees. Using counterfactual explanations for model reconstruction has received limited attention, with the notable exceptions of Aïvodji et al. [2020] and Wang et al. [2022]. Aïvodji et al. [2020] suggest using counterfactuals as ordinary labeled examples while training the surrogate model, leading to decision boundary shifts, particularly for unbalanced query datasets (one-sided counterfactuals). Wang et al. [2022] introduces a strategy of mitigating this issue by further querying for the counterfactual of the counterfactual. However, both these methods require the system to provide counterfactuals for queries from both sides of the decision boundary. Nevertheless, a user with a favorable decision may not usually require a counterfactual explanation, and hence a system providing one-sided counterfactuals might be more common, wherein lies our significance. While model reconstruction (without counterfactuals) has received interest from a theoretical perspective [Tramèr et al., 2016, Papernot et al., 2017, Milli et al., 2019], model reconstruction involving counterfactual explanations lack such a theoretical understanding. Our work theoretically analyzes model reconstruction using polytope theory and proposes novel strategies thereof, also addressing the decision-boundary shift issue.

## 2  Preliminaries

**Notations:** We consider binary classification models $m$ that take an input value $\boldsymbol{x} \in \mathbb{R}^d$ and output a probability between 0 and 1. The final predicted class is denoted by $\lfloor m(\boldsymbol{x}) \rceil \in \{0, 1\}$ which is obtained by thresholding the output probability at 0.5 as follows: $\lfloor m(\boldsymbol{x}) \rceil = \mathbb{1}[m(\boldsymbol{x}) \geq 0.5]$ where $\mathbb{1}[\cdot]$ denotes the indicator function. Throughout the paper, we denote the output probability by $m(\boldsymbol{x})$ and the corresponding thresholded output by $\lfloor m(\boldsymbol{x}) \rceil$. Consequently, the decision boundary of the model $m$ is the $(d-1)$-dimensional hypersurface (generalization of surface in higher dimensions; see Definition 2.5) in the input space, given by $\partial \mathbb{M} = \{\boldsymbol{x} : m(\boldsymbol{x}) = 0.5\}$. We call the region where $\lfloor m(\boldsymbol{x}) \rceil = 1$ as the *favorable region* and the region where $\lfloor m(\boldsymbol{x}) \rceil = 0$ as the *unfavorable region*. We always state the convexity/concavity of the decision boundary with respect to the favorable region (i.e., the decision boundary is convex if the set $\mathbb{M} = \{\boldsymbol{x} \in \mathbb{R}^d : \lfloor m(\boldsymbol{x}) \rceil = 1\}$ is convex). We assume that upon knowing the range of values for each feature, the $d-$dimensional input space can be normalized so that the inputs lie within the set $[0, 1]^d$ (the $d-$dimensional unit hypercube), as is common in literature [Liu et al., 2020, Tramèr et al., 2016, Hamman et al., 2023, Black et al., 2022]. We let $g_m$ denote the counterfactual generating mechanism corresponding to the model $m$.

**Definition 2.1** (Counterfactual Generating Mechanism). Given a cost function $c : [0, 1]^d \times [0, 1]^d \to \mathbb{R}_0^+$ for measuring the quality of a counterfactual, and a model $m$, the corresponding counterfactual generating mechanism is the mapping $g_m : [0, 1]^d \to [0, 1]^d$ specified as follows: $g_m(\boldsymbol{x}) = \arg \min_{\boldsymbol{w} \in [0,1]^d} c(\boldsymbol{x}, \boldsymbol{w})$, such that $\lfloor m(\boldsymbol{x}) \rceil \neq \lfloor m(\boldsymbol{w}) \rceil$.

The cost $c(\boldsymbol{x}, \boldsymbol{w})$ is selected based on specific desirable criteria, e.g., $c(\boldsymbol{x}, \boldsymbol{w}) = ||\boldsymbol{x} - \boldsymbol{w}||_p$, with $|| \cdot ||_p$ denoting the $L_p$-norm. Specifically, $p = 2$ leads to the following definition of the *closest counterfactual* [Wachter et al., 2017, Laugel et al., 2017, Mothilal et al., 2020].

**Definition 2.2** (Closest Counterfactual). When $c(\boldsymbol{x}, \boldsymbol{w}) \equiv ||\boldsymbol{x} - \boldsymbol{w}||_2$, the resulting counterfactual generated using $g_m$ as per Definition 2.1 is called the closest counterfactual.

Given a model $m$ and a counterfactual generating method $g_m$, we define the inverse counterfactual region $\mathbb{G}$ for a subset $\mathbb{H} \subseteq [0, 1]^d$ to be the region whose counterfactuals under $g_m$ fall in $\mathbb{H}$.

**Definition 2.3** (Inverse Counterfactual Region). The inverse counterfactual region $\mathbb{G}_{m,g_m}$ of $\mathbb{H} \subseteq [0, 1]^d$ is the the region defined as: $\mathbb{G}_{m,g_m}(\mathbb{H}) = \{\boldsymbol{x} \in [0, 1]^d : g_m(\boldsymbol{x}) \in \mathbb{H}\}$.

**Problem Setting:** Our problem setting involves a target model $m$ which is pre-trained and assumed to be hosted on a MLaaS platform (see Fig. 2). Any user can query it with a set of input instances $\mathbb{D} \subseteq [0,1]^d$ (also called counterfactual queries) and will be provided with the set of predictions, i.e., $\{\lfloor m(\boldsymbol{x}) \rceil : \boldsymbol{x} \in \mathbb{D}\}$, and a set of *one-sided* counterfactuals for the instances whose predicted class is 0, i.e., $\{g_m(\boldsymbol{x}) : \boldsymbol{x} \in \mathbb{D}, \lfloor m(\boldsymbol{x}) \rceil = 0\}$. Note that,

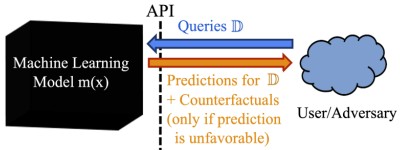

Figure 2: Problem setting

by the definition of a counterfactual, $\lfloor m(g_m(\boldsymbol{x})) \rceil = 1$ for all $\boldsymbol{x}$ with $\lfloor m(\boldsymbol{x}) \rceil = 0$. The **goal** of the user is to train a surrogate model to achieve a certain level of performance with as few queries as possible. In this work, we use *fidelity* as our performance metric for model reconstruction[1].

**Definition 2.4** (Fidelity [Aïvodji et al., 2020]). With respect to a given target model $m$ and a reference dataset $\mathbb{D}_{\text{ref}} \subseteq [0,1]^d$, the fidelity of a surrogate model $\tilde{m}$ is given by

$$\text{Fid}_{m,\mathbb{D}_{\text{ref}}}(\tilde{m}) = \frac{1}{|\mathbb{D}_{\text{ref}}|} \sum_{\boldsymbol{x} \in \mathbb{D}_{\text{ref}}} \mathbb{1}\left[\lfloor m(\boldsymbol{x}) \rceil = \lfloor \tilde{m}(\boldsymbol{x}) \rceil\right].$$

**Background on Geometry of Decision Boundaries:** Our theoretical analysis employs arguments based on the geometry of the involved models' decision boundaries. We assume the decision boundaries are hypersurfaces. A hypersurface is a generalization of a surface into higher dimensions, e.g., a line or a curve in a 2-dimensional space, a surface in a 3-dimensional space, etc.

**Definition 2.5** (Hypersurface, Lee [2009]). A hypersurface is a $(d-1)$-dimensional sub-manifold embedded in $\mathbb{R}^d$, which can also be denoted by a single implicit equation $\mathcal{S}(\boldsymbol{x}) = 0$ where $\boldsymbol{x} \in \mathbb{R}^d$.

We focus on the properties of hypersurfaces which are "touching" each other, as defined next.

**Definition 2.6** (Touching Hypersurfaces). Let $\mathcal{S}(\boldsymbol{x}) = 0$ and $\mathcal{T}(\boldsymbol{x}) = 0$ denote two differentiable hypersurfaces in $\mathbb{R}^d$. $\mathcal{S}(\boldsymbol{x}) = 0$ and $\mathcal{T}(\boldsymbol{x}) = 0$ are said to be touching each other at the point $\boldsymbol{w}$ if and only if $\mathcal{S}(\boldsymbol{w}) = \mathcal{T}(\boldsymbol{w}) = 0$, and there exists a non-empty neighborhood $\mathcal{B}_{\boldsymbol{w}}$ around $\boldsymbol{w}$, such that $\forall \boldsymbol{x} \in \mathcal{B}_{\boldsymbol{w}}$ with $\mathcal{S}(\boldsymbol{x}) = 0$ and $\boldsymbol{x} \neq \boldsymbol{w}$, only one of $\mathcal{T}(\boldsymbol{x}) > 0$ or $\mathcal{T}(\boldsymbol{x}) < 0$ holds. (i.e., within $\mathcal{B}_{\boldsymbol{w}}$, $\mathcal{S}(\boldsymbol{x}) = 0$ and $\mathcal{T}(\boldsymbol{x}) = 0$ lie on the same side of each other).

Next, we show that touching hypersurfaces share a common tangent hyperplane at their point of contact. This result is instrumental in exploiting the closest counterfactuals in model reconstruction (proof in Appendix A.1).

**Lemma 2.7.** *Let $\mathcal{S}(\boldsymbol{x}) = 0$ and $\mathcal{T}(\boldsymbol{x}) = 0$ denote two differentiable hypersurfaces in $\mathbb{R}^d$, touching each other at point $\boldsymbol{w}$. Then, $\mathcal{S}(\boldsymbol{x}) = 0$ and $\mathcal{T}(\boldsymbol{x}) = 0$ have a common tangent hyperplane at $\boldsymbol{w}$.*

## 3 Main Results

We provide a set of theoretical guarantees on query size of model reconstruction that progressively build on top of each other, starting from stronger guarantees in a somewhat restricted setting towards more approximate guarantees in broadly applicable settings. We discuss important intuitions that can be inferred from the results. Finally, we propose a model reconstruction strategy that is executable in practice, which is empirically evaluated in Section 4.

### 3.1 Convex decision boundaries and closest counterfactuals

We start out with demonstrating how the closest counterfactuals provide a linear approximation of *any* decision boundary, around the corresponding counterfactual. Prior work [Yadav et al., 2023] shows that for linear models, the line joining a query instance $\boldsymbol{x}$ and the closest counterfactual $\boldsymbol{w}(= g_m(\boldsymbol{x}))$ is perpendicular to the linear decision boundary. We generalize this observation to any differentiable decision boundary, not necessarily linear.

**Lemma 3.1.** *Let $\mathcal{S}$ denote the decision boundary of a classifier and $\boldsymbol{x} \in [0,1]^d$ be any point that is not on $\mathcal{S}$. Then, the line joining $\boldsymbol{x}$ and its closest counterfactual $\boldsymbol{w}$ is perpendicular to $\mathcal{S}$ at $\boldsymbol{w}$.*

---

[1]Performance can be evaluated using accuracy or fidelity [Jagielski et al., 2020]. Accuracy is a measure of how well the surrogate model can predict the true labels over the data manifold of interest. While attacks based on both measures have been proposed in literature, fidelity-based attacks have been deemed more useful as a first step in designing future attacks [Jagielski et al., 2020, Papernot et al., 2017].

The proof follows by showing that the $d$-dimensional ball with radius $||\boldsymbol{x}-\boldsymbol{w}||_2$ touches (as in Definition 2.6) $\mathcal{S}$ at $\boldsymbol{w}$, and invoking Lemma 2.7. For details see Appendix A.1. As a direct consequence of Lemma 3.1, a user may query the system and calculate tangent hyperplanes of the decision boundary drawn at the closest counterfactuals. This leads to a linear approximation of the decision boundary at the closest counterfactuals (see Fig. 3).

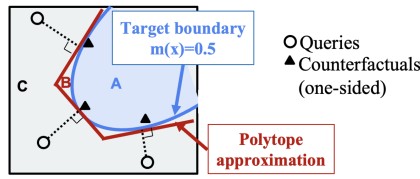

Figure 3: Polytope approximation of a convex decision boundary using the closest counterfactuals.

If the decision boundary is convex, such an approximation will provide a set of supporting hyperplanes. *The intersection of these supporting hyperplanes will provide a circumscribing convex polytope approximation of the decision boundary.* We show that the average fidelity of such an approximation, evaluated over uniformly distributed input instances, tends to 1 when the number of queries is large.

**Theorem 3.2.** *Let $m$ be the target binary classifier whose decision boundary is convex (i.e., the set $\{\boldsymbol{x} \in [0,1]^d : \lfloor m(\boldsymbol{x}) \rceil = 1\}$ is convex) and has a continuous second derivative. Denote by $\tilde{M}_n$, the convex polytope approximation of $m$ constructed with $n$ supporting hyperplanes obtained through i.i.d. counterfactual queries. Assume that the fidelity is evaluated with respect to $\mathbb{D}_{ref}$ which is uniformly distributed over $[0,1]^d$. Then, when $n \to \infty$ the expected fidelity of $\tilde{M}_n$ with respect to $m$ is given by*

$$\mathbb{E}\left[\mathrm{Fid}_{m,\mathbb{D}_{ref}}(\tilde{M}_n)\right] = 1 - \epsilon \tag{1}$$

*where $\epsilon \sim \mathcal{O}\left(n^{-\frac{2}{d-1}}\right)$ and the expectation is over both $\tilde{M}_n$ and $\mathbb{D}_{ref}$.*

Theorem 3.2 provides an exact relationship between the expected fidelity and number of queries. The proof utilizes a result from random polytope theory [Böröczky Jr and Reitzner, 2004] which provides a complexity bound on volume-approximating smooth convex sets by convex polytopes. The proof involves observing that the volume of the overlapping decision regions of $m$ and $\tilde{M}_n$ (for example, regions A and C in Fig. 3) translates to the expected fidelity when evaluated under a uniformly distributed $\mathbb{D}_{\mathrm{ref}}$. Appendix A.2 provides the detailed steps.

*Remark* 3.3 (Relaxing the Convexity Assumption). This strategy of linear approximations can also be extended to a concave decision boundary since the closest counterfactual will always lead to a tangent hyperplane irrespective of convexity. Now the rejected region can be seen as the intersection of these half-spaces (Lemma 3.1 does not assume convexity). However, it is worth noting that approximating a concave decision boundary is, in general, more difficult than approximating a convex region. To obtain equally-spaced out tangent hyperplanes on the decision boundary, a concave region will require a much denser set of query points (see Fig. 4) due to the inverse effect of length contraction discussed in Aleksandrov [1967, Chapter III Lemma 2]. *Deriving similar theoretical guarantees for a decision boundary which is neither convex nor concave is much more challenging as the decision regions can no longer be approximated as intersections of half-spaces.* The assumption of convex decision boundaries may only be satisfied under limited scenarios such as input-convex neural networks [Amos et al., 2017]. However, we observe that this limitation can be circumvented in case of ReLU networks to arrive at a probabilistic guarantee as discussed next.

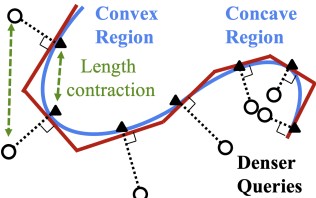

Figure 4: Approximating a concave region needs denser queries w.r.t. a convex region.

## 3.2 ReLU networks and closest counterfactuals

Rectified Linear Units (ReLU) are one of the most used activation functions in neural networks [Zeiler et al., 2013, Maas et al., 2013, Ronneberger et al., 2015, He et al., 2016]. A deep neural network that uses ReLU activations can be represented as a Continuous Piece-Wise Linear (CPWL) function [Chen et al., 2022, Hanin and Rolnick, 2019]. A CPWL function comprises of a union of linear functions over a partition of the domain. Definition 3.4 below provides a precise characterization.

**Definition 3.4** (Continuous Piece-Wise Linear (CPWL) Function [Chen et al., 2022]). A function $\ell : \mathbb{R}^d \to \mathbb{R}$ is said to be continuous piece-wise linear if and only if

1. There exists a finite set of closed subsets of $\mathbb{R}^d$, denoted as $\{\mathbb{U}_i\}_{i=1,2,\ldots,q}$ such that $\cup_{i=1}^q \mathbb{U}_i = \mathbb{R}^n$

2. $\ell(\boldsymbol{x})$ is affine over each $\mathbb{U}_i$ i.e., over each $\mathbb{U}_i$, $\ell(\boldsymbol{x}) = \ell_i(x) = \boldsymbol{a}_i^T \boldsymbol{x} + b_i$ with $\boldsymbol{a}_i \in \mathbb{R}^d$, $b_i \in \mathbb{R}$.

This definition can be readily applied to the models of our interest, of which the domain is the unit hypercube $[0,1]^d$. A neural network with ReLU activations can be used as a classifier by appending a Sigmoid activation $\sigma(z) = \frac{1}{1+e^{-z}}$ to the final output. We denote such a classifier by $m(\boldsymbol{x}) = \sigma(\ell(\boldsymbol{x}))$ where $\ell(\boldsymbol{x})$ is CPWL. It has been observed that the number of linear pieces $q$ of a trained ReLU network is generally way below the theoretically allowed maximum [Hanin and Rolnick, 2019].

We first show that the decision boundaries of such CPWL functions are collections of polytopes (not necessarily convex). The proof of Lemma 3.5 is deferred to Appendix A.3.

**Lemma 3.5.** *Let $m(\boldsymbol{x}) = \sigma(\ell(\boldsymbol{x}))$ be a ReLU classifier, where $\ell(\boldsymbol{x})$ is CPWL and $\sigma(.)$ is the Sigmoid function. Then, the decision boundary $\partial\mathbb{M} = \{\boldsymbol{x} \in [0,1]^d : m(\boldsymbol{x}) = 0.5\}$ is a collection of (possibly non-convex) polytopes in $[0,1]^d$, when considered along with the boundaries of the unit hypercube.*

Next we will analyze the probability of successful model reconstruction using counterfactuals. Consider a uniform grid $\mathcal{N}_\epsilon$ over the unit hypercube $[0,1]^d$, where each cell is a smaller hypercube with side length $\epsilon$ (see Fig. 5). For this analysis, we make an assumption: *If a cell contains a part of the decision boundary, then that part is completely linear (affine) within that small cell* [2].

Now, as the decision boundary becomes linear for each small cell that it passes through, having just one closest counterfactual in each such cell is sufficient to get the decision boundary in that cell (recall Lemma 3.1). We formalize this intuition in Theorem 3.6 to obtain a probabilistic guarantee on the success of model reconstruction. A proof is presented in Appendix A.4.

**Theorem 3.6.** *Let $m$ be a target binary classifier with ReLU activations. Let $k(\epsilon)$ be the number of cells through which the decision boundary passes. Define $\{\mathbb{H}_i\}_{i=1,\ldots,k(\epsilon)}$ to be the collection of affine pieces of the decision boundary within each decision boundary cell where each $\mathbb{H}_i$ is an open set. Let $v_i(\epsilon) = V(\mathbb{G}_{m,g_m}(\mathbb{H}_i))$ where $V(.)$ is the $d-$dimensional volume (i.e., the Lebesgue measure) and $\mathbb{G}_{m,g_m}(.)$ is the inverse counterfactual region w.r.t. $m$ and the closest counterfactual generator $g_m$. Then the probability of successful reconstruction with counterfactual queries distributed uniformly over $[0,1]^d$ is lower-bounded as*

$$\mathbb{P}\left[Reconstruction\right] \geq 1 - k(\epsilon)(1 - v^*(\epsilon))^n \qquad (2)$$

*where $v^*(\epsilon) = \min_{i=1,\ldots,k(\epsilon)} v_i(\epsilon)$ and $n$ is the number of queries.*

*Remark* 3.7. Here $k(\epsilon)$ and $v^*(\epsilon)$ depend only on the nature of the model being reconstructed and are independent of the number of queries $n$. The value of $k(\epsilon)$ roughly grows with the surface area of the decision boundary (e.g., length when input is 2D), showing that models with more convoluted decision boundaries might need more queries for reconstruction. Generally, $k(\epsilon)$ lies within the interval $\frac{A(\partial\mathbb{M})}{\sqrt{2}\epsilon^{d-1}} \leq k(\epsilon) \leq \frac{1}{\epsilon^d}$ where $A(.)$ denotes the surface area in $d-$dimensional space. The lower bound is due to the fact that the area of any slice of the unit hypercube being at-most $\sqrt{2}$ [Ball, 1986]. Upper bound is reached when the decision boundary traverses through all the cells in the grid which is less likely in practice. When the model complexity increases, we get a larger $k(\epsilon)$ as well as a smaller $v^*(\epsilon)$, requiring a higher number of queries to achieve similar probabilities of success.

Figure 5: $\mathcal{N}_\epsilon$ grid and inverse counterfactual regions. Thick solid lines indicate the decision boundary pieces ($\mathbb{H}_i$'s). White color depicts the accepted region. Pale-colored are the inverse counterfactual regions of the $\mathbb{H}_i$'s with the matching color. In this case $k(\epsilon) = 7$ and $v^*(\epsilon)$ is the area of lower amber region.

**Corollary 3.8** (Linear Models). *For linear models with one-sided counterfactuals, $\mathbb{P}\left[Reconstruction\right] = 1 - (1-v)^n$ where $v$ is the volume of the unfavorable region. However, with two-sided counterfactuals, $\mathbb{P}\left[Reconstruction\right] = 1$ with just one single query.*

---

[2]This is violated only for the cells containing parts of the edges of the decision boundary. However, we may assume that $\epsilon$ is small enough so that the total number of such cells is negligible compared to the total cells containing the decision boundary. Generally, $\epsilon$ can be made sufficiently small such that for most of the cells, if a cell contains a part of the decision boundary then that part is affine within the cell.

This result mathematically demonstrates that allowing counterfactuals from both accepted and rejected regions (as in Aïvodji et al. [2020], Wang et al. [2022]) is easier for model reconstruction, when compared to the one-sided case. It effectively increases each $v_i(\epsilon)$ (volume of the inverse counterfactual region). As everything else remains unaffected, for a given $n$, $\mathbb{P}[\text{Reconstruction}]$ is higher when counterfactuals from both regions are available. For a linear model, this translates to a guaranteed reconstruction with a single query since $v = 1$.

However, we note that all of the aforementioned analysis relies on the closest counterfactual which can be challenging to generate. Practical counterfactual generating mechanisms usually provide counterfactuals that are reasonably close but may not be exactly the closest. This motivates us to now propose a more general strategy assuming local Lipschitz continuity of the models involved.

### 3.3  Beyond closest counterfactuals

Lipschitz continuity, a property that is often encountered in related works [Bartlett et al., 2017, Gouk et al., 2021, Pauli et al., 2021, Hamman et al., 2023, 2024, Liu et al., 2020, Marques-Silva et al., 2021], demands the model output does not change too fast. Usually, a smaller Lipschitz constant is indicative of a higher generalizability of a model [Gouk et al., 2021, Pauli et al., 2021].

**Definition 3.9** (Local Lipschitz Continuity). A model $m$ is said to be locally Lipschitz continuous if for every $\boldsymbol{x}_1 \in [0,1]^d$ there exists a neighborhood $\mathbb{B}_{\boldsymbol{x}_1} \subseteq [0,1]^d$ around $\boldsymbol{x}_1$ such that for all $\boldsymbol{x}_2 \in \mathbb{B}_{\boldsymbol{x}_1}, |m(\boldsymbol{x}_1) - m(\boldsymbol{x}_2)| \leq \gamma ||\boldsymbol{x}_1 - \boldsymbol{x}_2||_2$ for some $\gamma \in \mathbb{R}_0^+$.

For analyzing model reconstruction under local-Lipschitz assumptions, we consider the difference of the model output probabilities (before thresholding) as a measure of similarity between the target and surrogate models because it would force the decision boundaries to overlap. The proposed strategy is motivated from the following observation: the difference of two models' output probabilities corresponding to a given input instance $\boldsymbol{x}$ can be bounded by having another point with matching outputs in the affinity of the instance $\boldsymbol{x}$. This observation is formally stated in Theorem 3.10. See Appendix A.5 for a proof.

**Theorem 3.10.** *Let the target $m$ and surrogate $\tilde{m}$ be ReLU classifiers such that $m(\boldsymbol{w}) = \tilde{m}(\boldsymbol{w})$ for every counterfactual $\boldsymbol{w}$. For any point $\boldsymbol{x}$ that lies in a decision boundary cell, $|\tilde{m}(\boldsymbol{x}) - m(\boldsymbol{x})| \leq \sqrt{d}(\gamma_m + \gamma_{\tilde{m}})\epsilon$ holds with probability $p \geq 1 - k(\epsilon)(1 - v^*(\epsilon))^n$.*

Note that within each decision boundary cell, models are affine and hence locally Lipschitz for some $\gamma_m, \gamma_{\tilde{m}} \in \mathbb{R}_0^+$. Local Lipschitz property assures that the approximation is quite close ($\gamma_m, \gamma_{\tilde{m}}$ are small) except over a few small ill-behaved regions of the decision boundary. This result can be extended to any locally Lipschitz pair of models as stated in following corollary.

**Corollary 3.11.** *Suppose the target $m$ and surrogate $\tilde{m}$ are locally Lipschitz (not necessarily ReLU) such that $m(\boldsymbol{w}) = \tilde{m}(\boldsymbol{w})$ for every counterfactual $\boldsymbol{w}$. Assume the counterfactuals are well-spaced out and forms a $\delta$-cover over the decision boundary. Then $|\tilde{m}(\boldsymbol{x}) - m(\boldsymbol{x})| \leq (\gamma_m + \gamma_{\tilde{m}})\delta$, over the target decision boundary.*

Theorem 3.10 provides the motivation for a novel model reconstruction strategy. Let $\boldsymbol{w}$ be a counterfactual. Recall that $\partial \mathbb{M}$ denotes the decision boundary of $m$. As implied by the theorem, for any $\boldsymbol{x} \in \partial \mathbb{M}$, the deviation of the surrogate model output from the target model output can be bounded above by $\sqrt{d}(\gamma_m + \gamma_{\tilde{m}})\epsilon$ given that all the counterfactuals satisfy $m(\boldsymbol{w}) = \tilde{m}(\boldsymbol{w})$. Knowing that $m(\boldsymbol{w}) = 0.5$, we may design a loss function which **clamps** $\tilde{m}(\boldsymbol{w})$ to be 0.5. *Consequently, with a sufficient number of well-spaced counterfactuals to cover $\partial \mathbb{M}$, we may achieve arbitrarily small $|\tilde{m}(\boldsymbol{x}) - m(\boldsymbol{x})|$ at the decision boundary of $m$ (Fig. 6).*

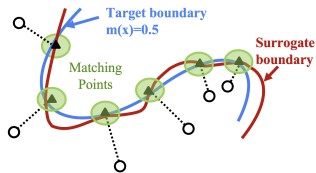

Figure 6: Rationale for Counterfactual Clamping Strategy.

However, we note that simply forcing $\tilde{m}(\boldsymbol{w})$ to be exactly equal to 0.5 is quite unstable since in practice the target model's output $m(\boldsymbol{w})$ is known to be close to 0.5, while being greater but may not be exactly equal. Thus, we propose a unique loss function for training the surrogate neural networks that does not penalize the counterfactuals that are already inside the favorable region of the surrogate model. For $0 < k \leq 1$, we define the Counterfactual Clamping loss function as

$$L_k(\tilde{m}(\boldsymbol{x}), y_{\boldsymbol{x}}) = \mathbb{1}\left[y_{\boldsymbol{x}} = 0.5, \tilde{m}(\boldsymbol{x}) \leq k\right]\left\{L(\tilde{m}(\boldsymbol{x}), k) - h(k)\right\} + \mathbb{1}\left[y_{\boldsymbol{x}} \neq 0.5\right]L(\tilde{m}(\boldsymbol{x}), y_{\boldsymbol{x}}). \quad (3)$$

Here, $y_{\boldsymbol{x}}$ denotes the label assigned to the input instance $\boldsymbol{x}$ by the target model, received from the API. $L(\hat{y}, y)$ is the binary cross-entropy loss and $h(\cdot)$ denotes the binary entropy function. We assume that the counterfactuals are distinguishable from the ordinary instances, and assign them a label of $y_{\boldsymbol{x}} = 0.5$. The first term accounts for the counterfactuals, where they are assigned a non-zero loss if the surrogate model's prediction is below $k$. This term ensures $\tilde{m}(\boldsymbol{w}) \gtrsim k$ for the counterfactuals $\boldsymbol{w}$ while not penalizing the ones that lie farther inside the favorable region. The second term is the ordinary binary cross-entropy loss, which becomes non-zero only for ordinary query instances. Note that substituting $k = 1$ in $L_k(\tilde{m}(\boldsymbol{x}), y_{\boldsymbol{x}})$ yields the ordinary binary cross entropy loss. Algorithm 1 summarizes the proposed strategy.

It is noteworthy that this approach is different from the broad area of soft-label learning Nguyen et al. [2011a,b] in two major aspects: (i) the labels in our problem do not smoothly span the interval [0,1] – instead they are either 0, 1 or 0.5; (ii) labels of counterfactuals do not indicate a class probability; even though a label of 0.5 is assigned, there can be counterfactuals that are well within the surrogate decision boundary which should not be penalized. Nonetheless, we also perform ablation studies where we compare the performance of our proposed loss function with another potential loss which simply forces $\tilde{m}(\boldsymbol{w})$ to be exactly 0.5, inspired from soft-label learning (see Appendix B.2.10 for results).

---

**Algorithm 1** Counterfactual Clamping

**Require:** Attack dataset $\mathbb{D}_{\text{attack}}$, $k$ ($k \in (0, 1]$, usually 0.5), API for querying
**Ensure:** Trained surrogate model $\tilde{m}$
1: Initialize $\mathbb{A} = \{\}$
2: **for** $\boldsymbol{x} \in \mathbb{D}_{\text{attack}}$ **do**
3:     Query API with $\boldsymbol{x}$ to get $y_{\boldsymbol{x}}$ $\{y_{\boldsymbol{x}} \in \{0, 1\}\}$
4:     $\mathbb{A} \leftarrow \mathbb{A} \cup \{(\boldsymbol{x}, y_{\boldsymbol{x}}\}$
5:     **if** $y_{\boldsymbol{x}} = 0$ **then**
6:         Query API for counterfactual $\boldsymbol{w}$ of $\boldsymbol{x}$
7:         $\mathbb{A} \leftarrow \mathbb{A} \cup \{(\boldsymbol{w}, 0.5)\}$ {Assign $\boldsymbol{w}$ a label of 0.5}
8:     **end if**
9: **end for**
10: Train $\tilde{m}$ on $\mathbb{A}$ with $L_k(\tilde{m}(\boldsymbol{x}), y_{\boldsymbol{x}})$ as the loss
11: **return** $\tilde{m}$

---

Counterfactual Clamping overcomes two challenges beset in existing works; (i) the problem of decision boundary shift (particularly with one-sided counterfactuals) present in the method suggested by Aïvodji et al. [2020] and (ii) the need for counterfactuals from both sides of the decision boundary in the methods of Aïvodji et al. [2020] and Wang et al. [2022].

# 4 Experiments

We carry out a number of experiments to study the performance of our proposed strategy Counterfactual Clamping. We include some results here and provide further details in Appendix B.

All the classifiers are neural networks unless specified otherwise and their decision boundaries are *not necessarily convex*. The performance of our strategy is compared with the existing attack presented in Aïvodji et al. [2020] that we refer to as "Baseline", for the case of one-sided counterfactuals. As the initial counterfactual generating method, we use an implementation of the Minimum Cost Counterfactuals (MCCF) by Wachter et al. [2017].

*Performance metrics:* Fidelity is used for evaluating the agreement between the target and surrogate models. It is evaluated over both uniformly generated instances (denoted by $\mathbb{D}_{\text{uni}}$) and test data instances from the data manifold (denoted by $\mathbb{D}_{\text{test}}$) as the reference dataset $\mathbb{D}_{\text{ref}}$.

A summary of the experiments is provided below with additional details in Appendix B.

**(i) Visualizing the attack using synthetic data:** First, the effect of the proposed loss function in mitigating the decision boundary shift is observed over a 2-D synthetic dataset. Fig. 7 presents the results. In the figure, it is clearly visible that the Baseline model is affected by a decision boundary shift. In contrast, the CCA model's decision boundary closely approximates the target decision boundary. See Appendix B.2.1 for more details.

**(ii) Comparing performance over four real-world dataset:** We use four publicly available real-world datasets namely, Adult Income, COMPAS, DCCC, and HELOC (see Appendix B.1) for our experiments. Table 1 provides some of the results over four real-world datasets. We refer to Appendix B.2.2 (specifically Fig. 11) for additional results. In all cases, we observe that CCA has either better or similar fidelity as compared to Baseline.

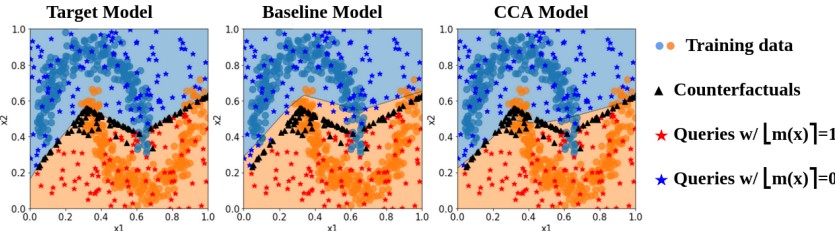

Figure 7: A 2-D demonstration of the proposed strategy. Orange and blue shades denote the favorable and unfavorable decision regions of each model. Circles denote the target model's training data.

Table 1: Average fidelity achieved with 400 queries on the real-world datasets over an ensemble of size 100. Target model has hidden layers with neurons (20,10). Model 0 is similar to the target model in architecture. Model 1 has hidden layers with neurons (20, 10, 5).

| Dataset | Architecture known (model 0) | | | | Architecture unknown (model 1) | | | |
|---|---|---|---|---|---|---|---|---|
| | $\mathbb{D}_{test}$ | | $\mathbb{D}_{uni}$ | | $\mathbb{D}_{test}$ | | $\mathbb{D}_{uni}$ | |
| | Base. | CCA | Base. | CCA | Base. | CCA | Base. | CCA |
| Adult In. | 91±3.2 | 94±3.2 | 84±3.2 | 91±3.2 | 91±4.5 | 94±3.2 | 84±3.2 | 90±3.2 |
| COMPAS | 92±3.2 | 96±2.0 | 94±1.7 | 96±2.0 | 91±8.9 | 96±3.2 | 94±2.0 | 94±8.9 |
| DCCC | 89±8.9 | 99±0.9 | 95±2.2 | 96±1.4 | 90±7.7 | 97±4.5 | 95±2.2 | 95±11.8 |
| HELOC | 91±4.7 | 96±2.2 | 92±2.8 | 94±2.4 | 90±7.4 | 95±5.5 | 91±3.3 | 93±3.2 |

**(iii) Studying effects of Lipschitz constants:** We study the connection between the target model's Lipschitz constant and the CCA performance. Target model's Lipschitz constant is controlled by changing the $L_2-$regularization coefficient, while keeping the surrogate models fixed. Results are presented in Fig. 15. Target models with a smaller Lipschitz constant are easier to extract. More details are provided in Appendix B.2.4.

**(iv) Studying different model architectures:** We also consider different surrogate model architectures spanning models that are more complex than the target model to much simpler ones. Results show that when sufficiently close to the target model in complexity, the surrogate architecture plays a little role on the performance. See Appendix B.2.5 for details. Furthermore, two situations are considered where the target model is not a neural network in Fig. 16 and Appendix B.2.8. In both scenarios, CCA surpasses the baseline.

**(v) Studying other counterfactual generating methods:** Effects of counterfactuals being sparse, actionable, realistic, and robust are observed. Sparse counterfactuals are generated by using $L_1-$norm as the cost function. Actionable counterfactuals are generated using DiCE [Mothilal et al., 2020] by defining a set of immutable features. Realistic counterfactuals (that lie on the data manifold) are generated by retrieving the 1-Nearest-Neighbor from the accepted side for a given query, as well as using the autoencoder-based method C-CHVAE [Pawelczyk et al., 2020]. Additionally, we generate robust counterfactuals using ROAR [Upadhyay et al., 2021]. We evaluate the attack performance on the HELOC dataset (Table 2). Moreover we observe the distribution of the counterfactuals generated using each method w.r.t. the target model's decision boundary using histograms (Fig. 8). Additional details are given in Appendix B.2.6.

**(vi) Comparison with DualCFX:** DualCFX proposed by Wang et al. [2022] is a strategy that utilizes the counterfactual of the counterfactuals to mitigate the decision boundary shift. We compare CCA with DualCFX in Table 6, Appendix B.2.7.

**(vii) Studying alternate loss functions:** We explore using binary cross-entropy loss function directly with labels 0, 1 and 0.5, in place of the proposed loss. However, experiments indicate that this scheme performs poorly when compared with the CCA loss (see Fig. 18 and Appendix B.2.10).

**(viii) Validating Theorem 3.2:** Empirical verification of the theorem is done through synthetic experiments, where the model has a spherical decision boundary since they are known to be more difficult for polytope approximation [Arya et al., 2012]. Fig. 20 presents a log-log plot comparing the theoretical and empirical query complexities for several dimensionality values $d$. The empirical approximation error decays faster than $n^{-2/(d-1)}$ as predicted by the theorem (see Appendix B.3).

Table 2: Fidelity achieved with different counterfactual generating methods on HELOC dataset. Target model has hidden layers with neurons (20, 30, 10). Surrogate model architecture is (10, 20).

| CF method | Fidelity over $\mathbb{D}_{test}$ | | | | Fidelity over $\mathbb{D}_{uni}$ | | | |
| | n=100 | | n=200 | | n=100 | | n=200 | |
| | Base. | CCA | Base. | CCA | Base. | CCA | Base. | CCA |
|---|---|---|---|---|---|---|---|---|
| MCCF L2-norm | 91 | 95 | 93 | 96 | 91 | 93 | 93 | 95 |
| MCCF L1-norm | 93 | 95 | 94 | 96 | 89 | 92 | 91 | 95 |
| DiCE Actionable | 93 | 94 | 95 | 95 | 90 | 91 | 93 | 94 |
| 1-Nearest-Neightbor | 93 | 95 | 94 | 96 | 93 | 93 | 94 | 95 |
| ROAR [Upadhyay et al., 2021] | 91 | 92 | 93 | 95 | 87 | 85 | 92 | 92 |
| C-CHVAE [Pawelczyk et al., 2020] | 77 | 80 | 78 | 82 | 90 | 89 | 85 | 78 |

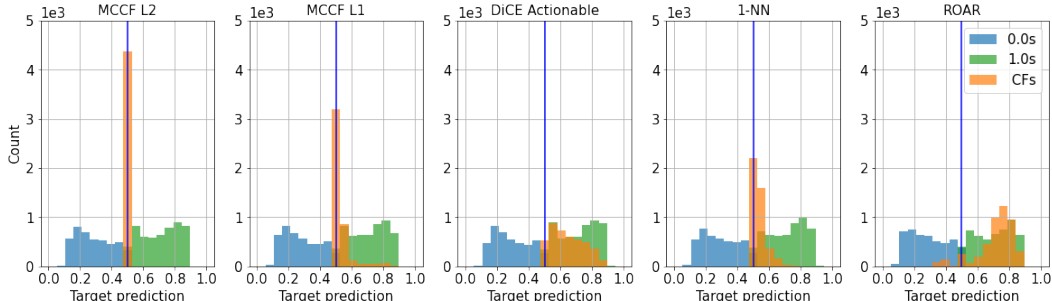

Figure 8: Histograms of the target model's predictions on different types of input instances. Counterfactual generating methods except MCCF with $L_2$ norm often generate counterfactuals that are farther inside the favorable region, hence having a target model prediction much greater than 0.5. We count all the query results across all the target models in the ensembles used to compute the average fidelities corresponding to each counterfactual generating method.

## 5   Conclusion

Our work provides novel insights that bridge explainability and privacy through a set of theoretical guarantees on model reconstruction using counterfactuals. We also propose a practical model reconstruction strategy based on the analysis. Experiments demonstrate a significant improvement in fidelity compared to the baseline method proposed in Aïvodji et al. [2020] for the case of one-sided counterfactuals. Results show that the attack performs well across different model types and counterfactual generating methods. Our findings also highlight an interesting connection between Lipschitz constant and vulnerability to model reconstruction.

**Limitations and Future Work:** Even though Theorem 3.6 provides important insights about the role of query size in model reconstruction, it lacks an exact characterization of $k(\epsilon)$ and $v_i(\epsilon)$. Moreover, local Lipschitz continuity might not be satisfied in some machine learning models such as decision trees. Any improvements along these lines can be avenues for future work. Utilizing techniques in active learning in conjunction with counterfactuals is another problem of interest. Extending the results of this work for multi-class classification scenarios can also be explored. The relationship between Lipschitz constant and vulnerability to model reconstruction may have implications for future work on generalization, robustness, etc. Another direction of future work is to design defenses for model reconstruction attacks, leveraging and building on strategies for robust counterfactuals [Dutta et al., 2022, Hamman et al., 2024, Upadhyay et al., 2021, Black et al., 2022, Jiang et al., 2023, Pawelczyk et al., 2020].

**Broader Impact:** We demonstrate that one-sided counterfactuals can be used for perfecting model reconstruction. While this can be beneficial in some cases, it also exposes a potential vulnerability in MLaaS platforms. Given the importance of counterfactuals in explaining model predictions, we hope our work will inspire countermeasures and defense strategies, paving the way toward secure and trustworthy machine learning systems.

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

# A    Proof of Theoretical Results

## A.1    Proof of Lemma 2.7 and Lemma 3.1

**Lemma 2.7.** *Let $\mathcal{S}(\boldsymbol{x}) = 0$ and $\mathcal{T}(\boldsymbol{x}) = 0$ denote two differentiable hypersurfaces in $\mathbb{R}^d$, touching each other at point $\boldsymbol{w}$. Then, $\mathcal{S}(\boldsymbol{x}) = 0$ and $\mathcal{T}(\boldsymbol{x}) = 0$ have a common tangent hyperplane at $\boldsymbol{w}$.*

*Proof.* From Definition 2.6, there exists a non-empty neighborhood $\mathcal{B}_{\boldsymbol{w}}$ around $\boldsymbol{w}$, such that $\forall \boldsymbol{x} \in \mathcal{B}_{\boldsymbol{w}}$ with $\mathcal{S}(\boldsymbol{x}) = 0$ and $\boldsymbol{x} \neq \boldsymbol{w}$, only one of $\mathcal{T}(\boldsymbol{x}) > 0$ or $\mathcal{T}(\boldsymbol{x}) < 0$ holds. Let $\boldsymbol{x} = (x_1, x_2, \ldots, x_d)$ and $\boldsymbol{x}_{[p]}$ denote $\boldsymbol{x}$ without $x_p$ for $1 \leq p \leq d$. Then, within the neighborhood $\mathcal{B}_{\boldsymbol{w}}$, we may re-parameterize $\mathcal{S}(\boldsymbol{x}) = 0$ as $x_p = S(\boldsymbol{x}_{[p]})$. Note that a similar re-parameterization denoted by $x_p = T(\boldsymbol{x}_{[p]})$ can be applied to $\mathcal{T}(\boldsymbol{x}) = 0$ as well. Let $\mathcal{A}_{\boldsymbol{w}} = \{\boldsymbol{x}_{[p]} : \boldsymbol{x} \in \mathcal{B}_{\boldsymbol{w}} \setminus \{\boldsymbol{w}\}\}$. From Definition 2.6, all $\boldsymbol{x} \in \mathcal{B}_{\boldsymbol{w}} \setminus \{\boldsymbol{w}\}$ satisfy only one of $\mathcal{T}(\boldsymbol{x}) < 0$ or $\mathcal{T}(\boldsymbol{x}) > 0$, and hence without loss of generality the re-parameterization of $\mathcal{T}(\boldsymbol{x}) = 0$ can be such that $S(\boldsymbol{x}_{[p]}) < T(\boldsymbol{x}_{[p]})$ holds for all $\boldsymbol{x}_{[p]} \in \mathcal{A}_{\boldsymbol{w}}$. Now, define $F(\boldsymbol{x}_{[p]}) \equiv T(\boldsymbol{x}_{[p]}) - S(\boldsymbol{x}_{[p]})$. Observe that $F(\boldsymbol{x}_{[p]})$ has a minimum at $\boldsymbol{w}$ and hence, $\nabla_{\boldsymbol{x}_{[p]}} F(\boldsymbol{w}_{[p]}) = 0$. Consequently, $\nabla_{\boldsymbol{x}_{[p]}} T(\boldsymbol{w}_{[p]}) = \nabla_{\boldsymbol{x}_{[p]}} S(\boldsymbol{w}_{[p]})$, which implies that the tangent hyperplanes to both hypersurfaces have the same gradient at $\boldsymbol{w}$. Proof concludes by observing that since both tangent hyperplanes go through $\boldsymbol{w}$, the two hypersurfaces should share a common tangent hyperplane at $\boldsymbol{w}$. $\qquad\square$

**Lemma 3.1.** *Let $\mathcal{S}$ denote the decision boundary of a classifier and $\boldsymbol{x} \in [0,1]^d$ be any point that is not on $\mathcal{S}$. Then, the line joining $\boldsymbol{x}$ and its closest counterfactual $\boldsymbol{w}$ is perpendicular to $\mathcal{S}$ at $\boldsymbol{w}$.*

*Proof.* The proof utilizes the following lemma.

**Lemma A.1.** *Consider the d-dimensional ball $\mathcal{C}_{\boldsymbol{x}}$ centered at $\boldsymbol{x}$, with $\boldsymbol{w}$ lying on its boundary (hence $\mathcal{C}_{\boldsymbol{x}}$ intersects $\mathcal{S}$ at $\boldsymbol{w}$). Then, $\mathcal{S}$ lies completely outside $\mathcal{C}_{\boldsymbol{x}}$.*

The proof of Lemma A.1 follows from the following contradiction. Assume a part of $\mathcal{S}$ lies within $\mathcal{C}_{\boldsymbol{x}}$. Then, points on the intersection of $\mathcal{S}$ and the interior of $\mathcal{C}_{\boldsymbol{x}}$ are closer to $\boldsymbol{x}$ than $\boldsymbol{w}$. Hence, $\boldsymbol{w}$ can no longer be the closest point to $\boldsymbol{x}$, on $\mathcal{S}$.

From Lemma A.1, $\mathcal{C}_{\boldsymbol{x}}$ is touching the curve $\mathcal{S}$ at $\boldsymbol{w}$, and hence, they share the same tangent hyperplane at $\boldsymbol{w}$ by Lemma 2.7. Now, observing that the line joining $\boldsymbol{w}$ and $\boldsymbol{x}$, being a radius of $\mathcal{C}_{\boldsymbol{x}}$, is the normal to the ball at $\boldsymbol{w}$ concludes the proof (see Fig. 9). $\qquad\square$

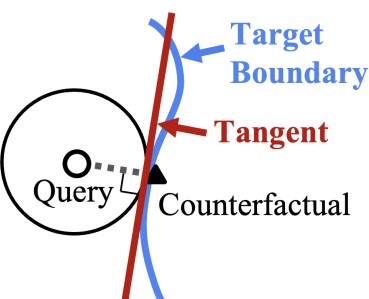

Figure 9: Line joining the query and its closest counterfactual is perpendicular to the decision boundary at the counterfactual. See Lemma 3.1 for details.

We present the following corollary as an additional observation resulting from Lemma A.1.

**Corollary A.2.** *Following Lemma A.1, it can be seen that all the points in the d-dimensional ball with $\boldsymbol{x}$ as the center and $\boldsymbol{w}$ on boundary lies on the same side of $\mathcal{S}$ as $\boldsymbol{x}$.*

## A.2 Proof of Theorem 3.2

**Theorem 3.2.** *Let $m$ be the target binary classifier whose decision boundary is convex (i.e., the set $\{\boldsymbol{x} \in [0,1]^d : \lfloor m(\boldsymbol{x}) \rceil = 1\}$ is convex) and has a continuous second derivative. Denote by $\tilde{M}_n$, the convex polytope approximation of $m$ constructed with $n$ supporting hyperplanes obtained through i.i.d. counterfactual queries. Assume that the fidelity is evaluated with respect to $\mathbb{D}_{ref}$ which is uniformly distributed over $[0,1]^d$. Then, when $n \to \infty$ the expected fidelity of $\tilde{M}_n$ with respect to $m$ is given by*

$$\mathbb{E}\left[\text{Fid}_{m,\mathbb{D}_{ref}}(\tilde{M}_n)\right] = 1 - \epsilon \tag{1}$$

*where $\epsilon \sim \mathcal{O}\left(n^{-\frac{2}{d-1}}\right)$ and the expectation is over both $\tilde{M}_n$ and $\mathbb{D}_{ref}$.*

*Proof.* We first have a look at Böröczky Jr and Reitzner [2004, Theorem 1 (restated as Theorem A.3 below)] from the polytope theory. Let $\mathbb{M}$ be a compact convex set with a second-order differentiable boundary denoted by $\partial\mathbb{M}$. Let $\boldsymbol{a}_1, \ldots, \boldsymbol{a}_n$ be $n$ randomly chosen points on $\partial\mathbb{M}$, distributed independently and identically according to a given density $d_{\partial\mathbb{M}}$. Denote by $H_+(\boldsymbol{a}_i)$ the supporting hyperplane of $\partial\mathbb{M}$ at $\boldsymbol{a}_i$. Assume $C$ to be a large enough hypercube which contains $\mathbb{M}$ in its interior.

Now, define

$$\tilde{\mathbb{M}}_n = \bigcap_{i=1}^{n} H_+(\boldsymbol{a}_i) \cap C \tag{4}$$

which is the polytope created by the intersection of all the supporting hyperplanes. The theorem characterizes the expected difference of the volumes of $\mathbb{M}$ and $\tilde{\mathbb{M}}_n$.

**Theorem A.3** (Random Polytope Approximation, [Böröczky Jr and Reitzner, 2004]). *For a convex compact set $\mathbb{M}$ with second-order differentiable $\partial\mathbb{M}$ and non-zero continuous density $d_{\partial\mathbb{M}}$,*

$$\mathbb{E}\left[V(\tilde{\mathbb{M}}_n) - V(\mathbb{M})\right] = \tau\left(\partial\mathbb{M}, d\right) n^{-\frac{2}{d-1}} + o\left(n^{-\frac{2}{d-1}}\right) \tag{5}$$

*as $n \to \infty$, where $V(\cdot)$ denotes the volume (i.e., the Lebesgue measure), and $\tau(\partial\mathbb{M}, d)$ is a constant that depends only on the boundary $\partial\mathbb{M}$ and the dimensionality $d$ of the space.*

Let $\mathbf{x}_i, i = 1, \ldots, n$ be $n$ i.i.d queries from the $\lfloor m(\mathbf{x}) \rceil = 0$ region of the target model. Then, their corresponding counterfactuals $g_m(\mathbf{x}_i)$ are also i.i.d. Furthermore, they lie on the decision boundary of $m$. Hence, we may arrive at the following result.

**Corollary A.4.** *Let $\mathbb{M} = \{\boldsymbol{x} \in [0,1]^d : \lfloor m(\boldsymbol{x}) \rceil = 1\}$ and $\tilde{\mathbb{M}}_n = \{\boldsymbol{x} \in [0,1]^d : \lfloor \tilde{M}_n(\boldsymbol{x}) \rceil = 1\}$. Then, by Theorem A.3,*

$$\mathbb{E}\left[V(\tilde{\mathbb{M}}_n) - V(\mathbb{M})\right] \sim \mathcal{O}\left(n^{-\frac{2}{d-1}}\right) \tag{6}$$

*when $n \to \infty$. Note that $\mathbb{M} \subseteq \tilde{\mathbb{M}}_n$ and hence, the left-hand side is always non-negative.*

From Definition 2.4, we may write

$$\mathbb{E}\left[\text{Fid}_{m,\mathbb{D}_{ref}}(\tilde{M}_n)\right]$$

$$= \mathbb{E}\left[\frac{1}{|\mathbb{D}_{ref}|} \sum_{\mathbf{x} \in \mathbb{D}_{ref}} \mathbb{E}\left[\mathbb{1}\left[\lfloor m(\mathbf{x}) \rceil = \lfloor \tilde{M}_n(\mathbf{x}) \rceil\right]\right] \Big| \mathbb{D}_{ref}\right] \tag{7}$$

$$= \frac{1}{|\mathbb{D}_{ref}|}\mathbb{E}\left[\sum_{\mathbf{x} \in \mathbb{D}_{ref}} \mathbb{P}\left[\lfloor m(\mathbf{x}) \rceil = \lfloor \tilde{M}_n(\mathbf{x}) \rceil \Big| \mathbf{x}\right]\right] \quad (\because \text{ query size is fixed}) \tag{8}$$

$$= \mathbb{P}\left[\lfloor m(\mathbf{x}) \rceil = \lfloor \tilde{M}_n(\mathbf{x}) \rceil\right] \quad (\because \mathbf{x}\text{'s are i.i.d.}) \tag{9}$$

$$= \int_{\mathcal{M}_n} \mathbb{P}\left[\lfloor m(\mathbf{x}) \rceil = \lfloor \tilde{M}_n(\mathbf{x}) \rceil \Big| \tilde{M}_n(\mathbf{x}) = \tilde{m}_n(\mathbf{x})\right] \mathbb{P}\left[\tilde{M}_n(\mathbf{x}) = \tilde{m}_n(\mathbf{x})\right] \mathrm{d}\tilde{m}_n \tag{10}$$

where $\mathcal{M}_n$ is the set of all possible $\tilde{m}_n$'s.

Now, by noting that

$$\mathbb{P}\left[\lfloor m(\mathbf{x})\rceil = \left\lfloor \tilde{M}_n(\mathbf{x})\right\rceil \Big| \tilde{M}_n(\mathbf{x}) = \tilde{m}_n(\mathbf{x})\right] = 1 - \mathbb{P}\left[\lfloor m(\mathbf{x})\rceil \neq \left\lfloor \tilde{M}_n(\mathbf{x})\right\rceil \Big| \tilde{M}_n(\mathbf{x}) = \tilde{m}_n(\mathbf{x})\right],$$
(11)

we may obtain

$$\mathbb{E}\left[\mathrm{Fid}_{m,\mathbb{D}_{\mathrm{ref}}}(\tilde{M}_n)\right] = 1 - \int_{\mathcal{M}_n} \mathbb{P}\left[\lfloor m(\mathbf{x})\rceil \neq \left\lfloor \tilde{M}_n(\mathbf{x})\right\rceil \Big| \tilde{M}_n(\mathbf{x}) = \tilde{m}_n(\mathbf{x})\right]$$
$$\times \mathbb{P}\left[\tilde{M}_n(\mathbf{x}) = \tilde{m}_n(\mathbf{x})\right] \mathrm{d}\tilde{m}_n \quad (12)$$

$$= 1 - \int_{\mathcal{M}_n} \underbrace{\frac{V(\tilde{\mathbb{M}}_n) - V(\mathbb{M})}{\text{Total volume}}}_{=1 \text{ for unit hypercube}} \mathbb{P}\left[\tilde{M}_n(\mathbf{x}) = \tilde{m}_n(\mathbf{x})\right] \mathrm{d}\tilde{m}_n$$

$$(\because \mathbf{x}\text{'s are uniformly distributed}) \quad (13)$$

$$= 1 - \mathbb{E}\left[V(\tilde{\mathbb{M}}_n) - V(\mathbb{M})\right]. \quad (14)$$

The above result, in conjunction with Corollary A.4, concludes the proof. $\qquad\square$

## A.3 Proof of Lemma 3.5

**Lemma 3.5.** *Let $m(\boldsymbol{x}) = \sigma(\ell(\boldsymbol{x}))$ be a ReLU classifier, where $\ell(\boldsymbol{x})$ is CPWL and $\sigma(.)$ is the Sigmoid function. Then, the decision boundary $\partial\mathbb{M} = \{\boldsymbol{x} \in [0,1]^d : m(\boldsymbol{x}) = 0.5\}$ is a collection of (possibly non-convex) polytopes in $[0,1]^d$, when considered along with the boundaries of the unit hypercube.*

*Proof.* Consider the $i^{\mathrm{th}}$ piece $m_i(\boldsymbol{x})$ of the classifier defined over $\mathbb{U}_i$. A part of the decision boundary exists within $\mathbb{U}_i$ only if $\exists \boldsymbol{x} \in \mathbb{U}_i$ such that $m_i(\boldsymbol{x}) = 0.5$. When it is the case, at the decision boundary,

$$m(\boldsymbol{x}) = 0.5 \quad (15)$$

$$\iff \frac{1}{1 + e^{-\ell_i(\boldsymbol{x})}} = 0.5 \quad (16)$$

$$\iff e^{-\ell_i(\boldsymbol{x})} = 1 \quad (17)$$

$$\iff \ell_i(\boldsymbol{x}) = 0 \quad (18)$$

$$\iff \boldsymbol{a}_i^T \boldsymbol{x} + b_i = 0 \quad (19)$$

which represents a hyperplane restricted to $\mathbb{U}_i$. Moreover, the continuity of the $\ell(\boldsymbol{x})$ demands the decision boundary to be continuous across the boundaries of $\mathbb{U}_i$'s. This fact can be proved as follows:

Note that within each region $\mathbb{U}_i$, exactly one of the following three conditions holds:

(a) $\forall \boldsymbol{x} \in \mathbb{U}_i, \ell_i(\boldsymbol{x}) > 0 \quad \rightarrow \quad \mathbb{U}_i$ does not contain a part of the decision boundary

(b) $\forall \boldsymbol{x} \in \mathbb{U}_i, \ell_i(\boldsymbol{x}) < 0 \quad \rightarrow \quad \mathbb{U}_i$ does not contain a part of the decision boundary

(c) $\exists \boldsymbol{x} \in \mathbb{U}_i, \ell_i(\boldsymbol{x}) = 0 \quad \rightarrow \quad \mathbb{U}_i$ contains a part of the decision boundary

In case when (c) holds for some region $\mathbb{U}_i$, the decision boundary within $\mathbb{U}_i$ is affine and it extends from one point to another on the region boundary. Now let $\mathbb{U}_s$ and $\mathbb{U}_t, s, t \in \{1, \ldots, q\}, s \neq t$ be two adjacent regions sharing a boundary. Assume that $\mathbb{U}_s$ contains a portion of the decision boundary, which intersects with a part of the shared boundary between $\mathbb{U}_s$ and $\mathbb{U}_t$ (note that $\mathbb{U}_i$'s are closed and hence they include their boundaries). Let $\boldsymbol{x}_0$ be a point in the intersection of the decision boundary within $\mathbb{U}_s$ and the shared region boundary. Now, continuity of $\ell(\boldsymbol{x})$ at $\boldsymbol{x}_0$ requires $\ell_t(\boldsymbol{x}_0) = \ell_s(\boldsymbol{x}_0) = 0$. Hence, condition (c) holds for $\mathbb{U}_t$. Moreover, this holds for all the points in the said intersection. Therefore, if such a shared boundary exists between $\mathbb{U}_s$ and $\mathbb{U}_t$, then the decision boundary continues to $\mathbb{U}_t$. Applying the argument to all $\mathbb{U}_s - \mathbb{U}_t$ pairs show that each segment of the decision boundary either closes upon itself or ends at a boundary of the unit hypercube. Hence, when taken along with the boundaries of the unit hypercube, the decision boundary is a collection of polytopes. $\qquad\square$

## A.4 Proof of Theorem 3.6

**Theorem 3.6.** *Let $m$ be a target binary classifier with ReLU activations. Let $k(\epsilon)$ be the number of cells through which the decision boundary passes. Define $\{\mathbb{H}_i\}_{i=1,\ldots,k(\epsilon)}$ to be the collection of affine pieces of the decision boundary within each decision boundary cell where each $\mathbb{H}_i$ is an open set. Let $v_i(\epsilon) = V(\mathbb{G}_{m,g_m}(\mathbb{H}_i))$ where $V(.)$ is the $d-$dimensional volume (i.e., the Lebesgue measure) and $\mathbb{G}_{m,g_m}(.)$ is the inverse counterfactual region w.r.t. $m$ and the closest counterfactual generator $g_m$. Then the probability of successful reconstruction with counterfactual queries distributed uniformly over $[0,1]^d$ is lower-bounded as*

$$\mathbb{P}\left[Reconstruction\right] \geq 1 - k(\epsilon)(1 - v^*(\epsilon))^n \tag{2}$$

*where $v^*(\epsilon) = \min_{i=1,\ldots,k(\epsilon)} v_i(\epsilon)$ and $n$ is the number of queries.*

*Proof.* Note that

$$\mathbb{P}[\text{Reconstruction}] = \mathbb{P}[\text{There is a counterfactual in every decision boundary cell}] \tag{20}$$

$$= 1 - \mathbb{P}[\text{At least one decision boundary cell does not have a counterfactual}] \tag{21}$$

$$= 1 - \sum_{i=1}^{k(\epsilon)} \mathbb{P}[i^{\text{th}} \text{ decision boundary cell does not have a counterfactual}] \tag{22}$$

Let $\mathcal{M}_i$ denote the event "$i^{\text{th}}$decision boundary cell does not have a counterfactual". At the end of $n$ queries,

$$\mathbb{P}[\mathcal{M}_i] = \prod_{j=1}^{n} \underbrace{\mathbb{P}[j^{\text{th}}\text{query falling outside of } \mathbb{G}_{m,g_m}(\mathbb{H}_i)]}_{=1-v_i(\epsilon) \text{ for uniform queries}} \tag{23}$$

$$= (1 - v_i(\epsilon))^n. \tag{24}$$

Therefore,

$$\mathbb{P}[\text{Reconstruction}] = 1 - \sum_{i=1}^{k(\epsilon)} (1 - v_i(\epsilon))^n \tag{25}$$

$$\geq 1 - k(\epsilon)(1 - v^*(\epsilon))^n \quad \left(\because v_i(\epsilon) \geq v^*(\epsilon) = \min_j v_j(\epsilon)\right). \tag{26}$$

$\square$

## A.5 Proof of Theorem 3.10 and Corollary 3.11

**Theorem 3.10.** *Let the target $m$ and surrogate $\tilde{m}$ be ReLU classifiers such that $m(\boldsymbol{w}) = \tilde{m}(\boldsymbol{w})$ for every counterfactual $\boldsymbol{w}$. For any point $\boldsymbol{x}$ that lies in a decision boundary cell, $|\tilde{m}(\boldsymbol{x}) - m(\boldsymbol{x})| \leq \sqrt{d}(\gamma_m + \gamma_{\tilde{m}})\epsilon$ holds with probability $p \geq 1 - k(\epsilon)(1 - v^*(\epsilon))^n$.*

**Corollary 3.11.** *Suppose the target $m$ and surrogate $\tilde{m}$ are locally Lipschitz (not necessarily ReLU) such that $m(\boldsymbol{w}) = \tilde{m}(\boldsymbol{w})$ for every counterfactual $\boldsymbol{w}$. Assume the counterfactuals are well-spaced out and forms a $\delta$-cover over the decision boundary. Then $|\tilde{m}(\boldsymbol{x}) - m(\boldsymbol{x})| \leq (\gamma_m + \gamma_{\tilde{m}})\delta$, over the target decision boundary.*

*Proof.* Note that from Theorem 3.6, with probability $p \geq 1 - k(\epsilon)(1 - v^*(\epsilon))^n$ at least one counterfactual exists within each decision boundary cell. When this is the case, we have

$$|\tilde{m}(\boldsymbol{x}) - m(\boldsymbol{x})| = |\tilde{m}(\boldsymbol{x}) - \tilde{m}(\boldsymbol{w}) - (m(\boldsymbol{x}) - \tilde{m}(\boldsymbol{w}))| \tag{27}$$

$$= |\tilde{m}(\boldsymbol{x}) - \tilde{m}(\boldsymbol{w}) - (m(\boldsymbol{x}) - m(\boldsymbol{w}))| \tag{28}$$

$$\leq \underbrace{|\tilde{m}(\boldsymbol{x}) - \tilde{m}(\boldsymbol{w})|}_{\leq \gamma_{\tilde{m}}||\boldsymbol{x}-\boldsymbol{w}||_2} + \underbrace{|m(\boldsymbol{x}) - m(\boldsymbol{w})|}_{\leq \gamma_m||\boldsymbol{x}-\boldsymbol{w}||_2} \tag{29}$$

$$\leq (\gamma_m + \gamma_{\tilde{m}})||\boldsymbol{x} - \boldsymbol{w}||_2 \tag{30}$$

$$\leq \sqrt{d}(\gamma_m + \gamma_{\tilde{m}})\epsilon \tag{31}$$

where the first inequality is a result of applying the triangle inequality and the second follows from the definition of local Lipschitz continuity (Definition 3.9). The final inequality is due to the availability of a counterfactual within each decision boundary cell, which ensures $||\boldsymbol{x} - \boldsymbol{w}||_2 \leq \sqrt{d}\epsilon$. Corollary 3.11 follows directly from the second inequality, since the $\delta-$cover of $\boldsymbol{w}$'s ensure $||\boldsymbol{x} - \boldsymbol{w}||_2 \leq \delta$ $\square$

# B  Experimental Details and Additional Results

All the experiments were carried-out on two computers, one with a NVIDIA RTX A4500 GPU and the other with a NVIDIA RTX 3050 Mobile.

## B.1  Details of Real-World Datasets

We use four publicly available real-world tabular datasets (namely, Adult Income, COMPAS, DCCC, and HELOC) to evaluate the performance of the attack proposed in Section 3.3. The details of these datasets are as follows:

- Adult Income: The dataset is a 1994 census database with information such as educational level, marital status, age and annual income of individuals [Becker and Kohavi, 1996]. The target is to predict "income", which indicates whether the annual income of a given person exceeds \$50000 or not (i.e., $y = \mathbb{1}[\text{income} \geq 0.5]$). It contains 32561 instances in total (the training set), comprising of 24720 from $y = 0$ and 7841 from $y = 1$. To make the dataset class-wise balanced we randomly sample 7841 instances from class $y = 0$, giving a total effective size of 15682 instances. Each instance has 6 numerical features and 8 categorical features. During pre-processing, categorical features are encoded as integers. All the features are then normalized to the range $[0, 1]$.

- Home Equity Line of Credit (HELOC): This dataset contains information about customers who have requested a credit line as a percentage of home equity FICO [2018]. It contains 10459 instances with 23 numerical features each. Prediction target is "is_at_risk" which indicates whether a given customer would pay the loan in the future. Dataset is slightly unbalanced with class sizes of 5000 and 5459 for $y = 0$ and $y = 1$, respectively. Instead of using all 23 features, we use the following subset of 10 for our experiments; "estimate_of_risk", "net_fraction_of_revolving_burden", "percentage_of_legal_trades", "months_since_last_inquiry_not_recent", "months_since_last_trade", "percentage_trades_with_balance", "number_of_satisfactory_trades", "average_duration_of_resolution", "nr_total_trades", "nr_banks_with_high_ratio". All the features are normalized to lie in the range $[0, 1]$.

- Correctional Offender Management Profiling for Alternative Sanctions (COMPAS): This dataset has been used for investigating racial biases in a commercial algorithm used for evaluating reoffending risks of criminal defendants [Angwin et al., 2016]. It includes 6172 instances and 20 numerical features. The target variable is "is_recid". Class-wise counts are 3182 and 2990 for $y = 0$ and $y = 1$, respectively. All the features are normalized to the interval $[0, 1]$ during pre-processing.

- Default of Credit Card Clients (DCCC): The dataset includes information about credit card clients in Taiwan Yeh [2016]. The target is to predict whether a client will default on the credit or not, indicated by "default.payment.next.month". The dataset contains 30000 instances with 24 attributes each. Class-wise counts are 23364 from $y = 0$ and 6636 from $y = 1$. To alleviate the imbalance, we randomly select 6636 instances from $y = 0$ class, instead of using all the instances. Dataset has 3 categorical attributes, which we encode into integer values. All the attributes are normalized to $[0, 1]$ during pre-processing.

## B.2  Experiments on the attack proposed in Section 3.3

In this section, we provide details about our experimental setup with additional results. For convenience, we present the neural network model architectures by specifying the number of neurons in each hidden layer as a tuple, where the leftmost element corresponds to the layer next to the input; e.g.: a model specified as (20,30,10) has the following architecture:

Input $\rightarrow$ Dense(20, ReLU) $\rightarrow$ Dense(30, ReLU) $\rightarrow$ Dense(10, ReLU) $\rightarrow$ Output(Sigmoid)

Other specifications of the models, as detailed below, are similar across most of the experiments. Changes are specified specifically. The hidden layer activations are ReLU and the layer weights are $L_2-$regularized. The regularization coefficient is 0.001. Each model is trained for 200 epochs, with a batch size of 32.

Fidelity is evaluated over a uniformly sampled set of input instances (uniform data) as well as a held-out portion of the original data (test data). The experiments were carried out as follows:

1. Initialize the target model. Train using $\mathbb{D}_{\text{train}}$.
2. For $t = 1, 2, \ldots, T$ :
   (a) Sample $N \times t$ data points from the dataset to create $\mathbb{D}_{\text{attack}}$.
   (b) Carry-out the attack given in Algorithm 1 with $\mathbb{D}_{\text{attack}}$. Use $k = 1$ for "Baseline" models and $k = 0.5$ for "Proposed" models.
   (c) Record the fidelity over $\mathbb{D}_{\text{ref}}$ along with $t$.
3. Repeat steps 1 and 2 for $S$ number of times and calculate average fidelities for each $t$, across repetitions.

Based on the experiments of Aïvodji et al. [2020] and Wang et al. [2022], we select $T = 20, 50, 100$; $N = 20, 8, 4$ and $S = 100, 50$, in different experiments. We note that the exact numerical results are often variable due to the multiple random factors affecting the outcome such as the test-train-attack split, target and surrogate model initialization, and the randomness incorporated in the counterfactual generating methods. Nevertheless, the advantage of CCA over the baseline attack is observed across different realizations.

### B.2.1 Visualizing the attack using synthetic data

This experiment is conducted on a synthetic dataset which consists of 1000 samples generated using the `make_moons` function from the `sklearn` package. Features are normalized to the range $[0, 1]$ before feeding to the classifier. The target model has 4 hidden layers with the architecture (10, 20, 20, 10). The surrogate model is 3-layered with the architecture (10, 20, 20). Each model is trained for 100 epochs. Since the intention of this experiment is to demonstrate the functionality of the modified loss function given in (3), a large query of size 200 is used, instead of performing multiple small queries. Fig. 7 shows how the original model reconstruction proposed by Aïvodji et al. [2020] suffers from the boundary shift issue, while the model with the proposed loss function overcomes this problem. Fig. 10 illustrates the instances misclassified by the two surrogate models.

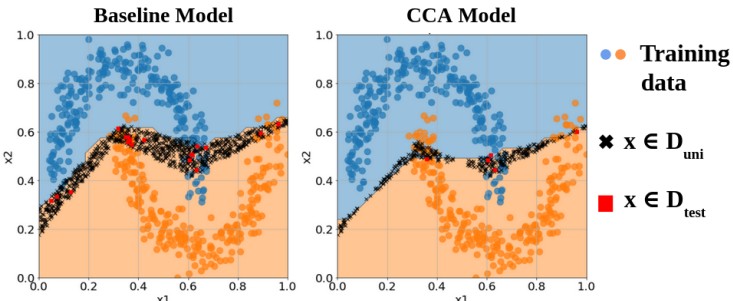

Figure 10: Misclassifications w.r.t. to the target model, over $\mathbb{D}_{\text{uni}}$ and $\mathbb{D}_{\text{test}}$ as the reference datasets for the 2-dimensional demonstration in Fig. 7. "Baseline" model causes a large number of misclassifications w.r.t. the "CCA" model.

### B.2.2 Comparing performance over four real-world dataset

We use a target model having 2 hidden layers with the architecture (20,10). Two surrogate model architectures, one exactly similar to the target architecture (model 0 - known architecture) and the other slightly different (model 1 - unknown architecture), are tested. Model 1 has 3 hidden layers with the architecture (20,10,5).

Fig. 11 illustrates the fidelities achieved by the two model architectures described above. Fig. 12 shows the corresponding variances of the fidelity values over 100 realizations. It can be observed that

the variances diminish as the query size grows, indicating more stable model reconstructions. Fig. 13 demonstrates the effect of the proposed loss function in mitigating the decision boundary shift issue.

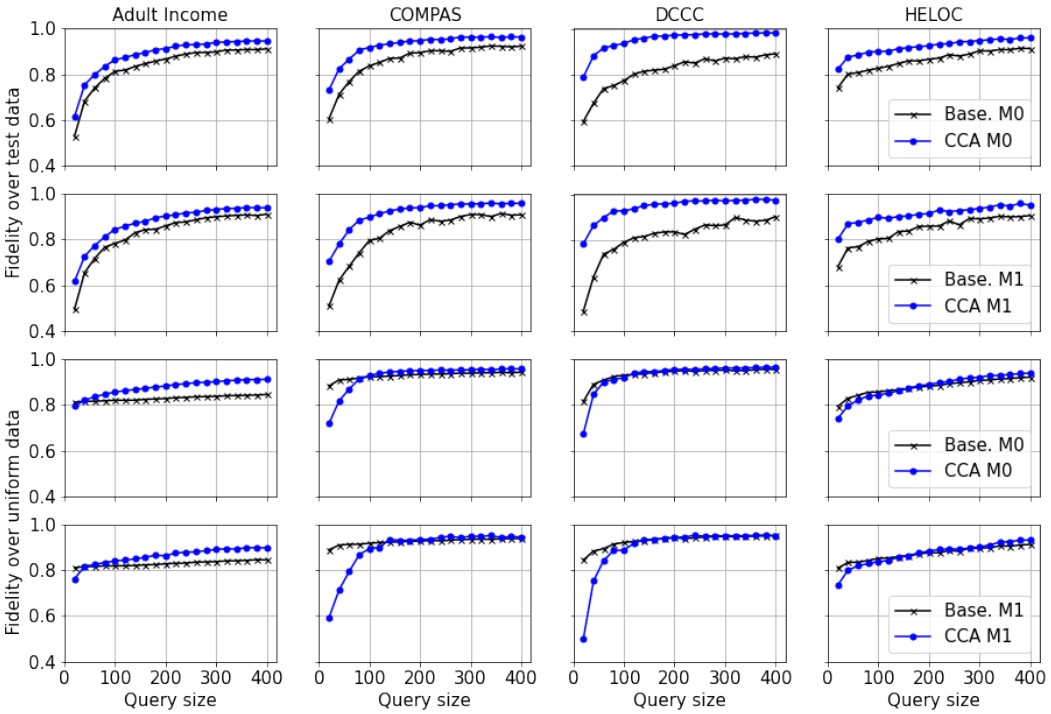

Figure 11: Fidelity for real-world datasets. Blue lines indicate "CCA" models. Black lines indicate "Baseline" models.

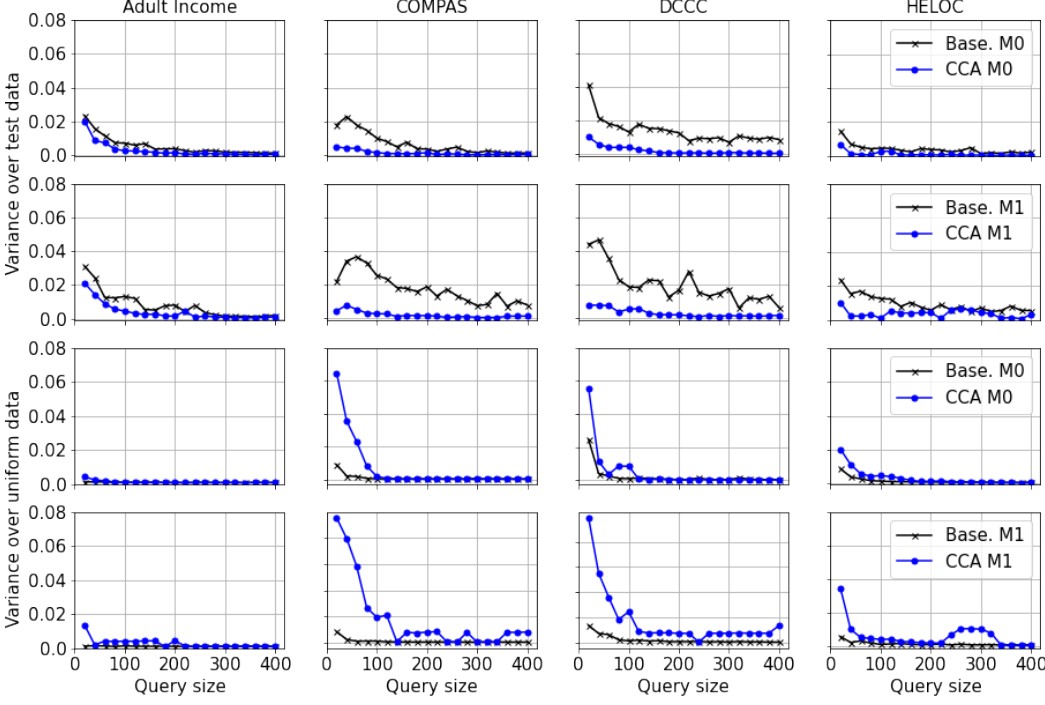

Figure 12: Variance of fidelity for real-world datasets. Blue lines indicate "CCA" models. Black lines indicate "Baseline" models.

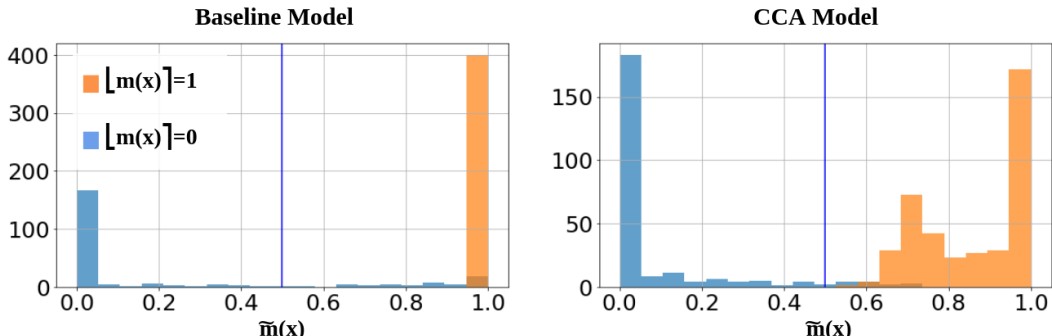

Figure 13: Histograms of probabilities predicted by "Baseline" and "CCA" models under the "Unknown Architecture" scenario (model 1) for the HELOC dataset. Note how the "Baseline" model provides predictions higher than 0.5 for a comparatively larger number of instances with $\lfloor m(\boldsymbol{x}) \rceil = 0$ due to the boundary shift issue. The clamping effect of the novel loss function is evident in the "CCA" model's histogram, where the decision boundary being held closer to the counterfactuals is causing the two prominent modes in the favorable region. The mode closer to 0.5 is due to counterfactuals and the mode closer to 1.0 is due to instances with $\lfloor m(\boldsymbol{x}) \rceil = 1$.

### B.2.3 Empirical and theoretical rates of convergence

Fig. 14 compares the rate of convergence of the empirical approximation error i.e., $1 - \mathbb{E}\left[\mathrm{Fid}_{m,\mathbb{D}_{\mathrm{ref}}}(\tilde{M}_n)\right]$ for two of the above experiments with the rate predicted by Theorem 3.2. Notice how the empirical error decays faster than $n^{-2/(d-1)}$.

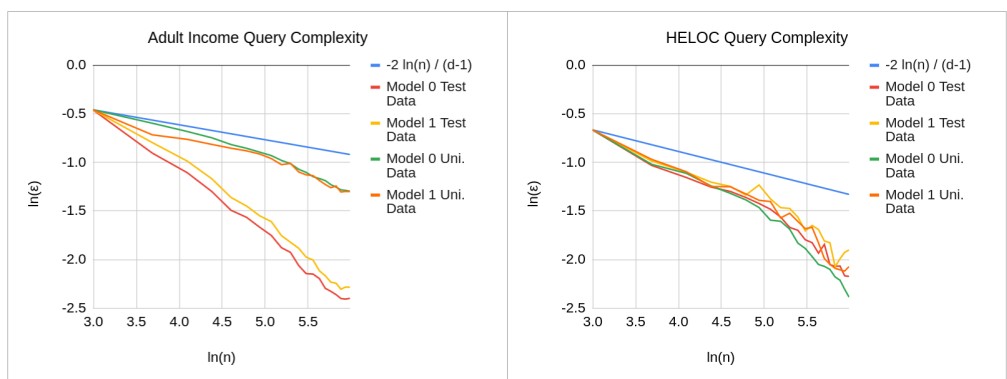

Figure 14: A comparison of the query complexity derived in Theorem 3.2 with the empirical query complexities obtained on the Adult Income and HELOC datasets. The graphs are on a log-log scale. We observe that the analytical query complexity is an upper bound for the empirical query complexities. All the graphs are recentered with an additive constant for presentational convenience. However, this does not affect the slope of the graph, which corresponds to the complexity.

### B.2.4 Studying effects of Lipschitz constants

For this experiment, we use a target model having 3 hidden layers with the architecture (20, 10, 5) and a surrogate model having 2 hidden layers with the architecture (20, 10). The surrogate model layers are $L_2$-regularized with a fixed regularization coefficient of 0.001. We achieve different Lipschitz constants for the target models by controlling their $L_2$-regularization coefficients during the target model training step. Following Gouk et al. [2021], we approximate the Lipschitz constant of target models by the product of the spectral norms of the weight matrices.

Fig. 15 illustrates the dependence of the attack performance on the Lipschitz constant of the target model. The results lead to the conclusion that target models with larger Lipschitz constants are more difficult to extract. This follows the insight provided by Theorem 3.10.

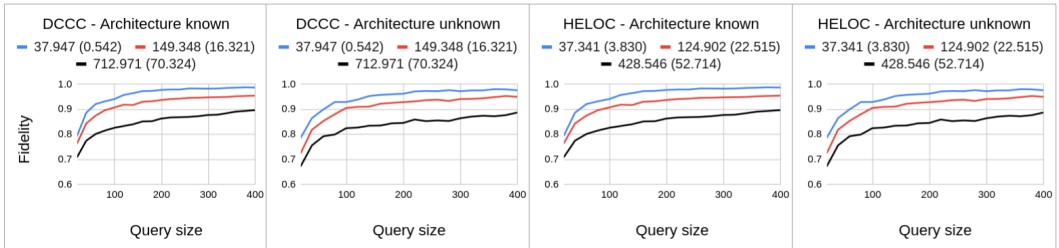

Figure 15: Dependence of fidelity on the target model's Lipschitz constant. The approximations of the Lipschitz constants are shown in the legend with standard deviations within brackets. Lipschitz constants are approximated as the product of the spectral norm of weight matrices in each model. With a higher Lipschitz constant, the fidelity achieved by a given number of queries tend to degrade.

### B.2.5 Studying different model architectures

We observe the effect of the model architectures on the attack performance over Adult Income, COMPAS and HELOC datasets. Tables 3, 4, and 5, respectively, present the results.

Table 3: Fidelity over $\mathbb{D}_{\text{test}}$ and $\mathbb{D}_{\text{uni}}$ for Adult Income dataset

| Target → | (20,10) | | | | (20,10,5) | | | | (20,20,10,5) | | | |
|---|---|---|---|---|---|---|---|---|---|---|---|---|
| $\mathbb{D}_{\text{test}}$ | n=100 | | n=200 | | n=100 | | n=200 | | n=100 | | n=200 | |
| Surrogate ↓ | Base. | CCA | Base. | CCA | Base. | CCA | Base. | CCA | Base. | CCA | Base. | CCA |
| (20,10) | 0.88 | 0.89 | 0.92 | 0.93 | 0.82 | 0.84 | 0.91 | 0.93 | 0.94 | 0.95 | 0.95 | 0.96 |
| (20,10,5) | 0.87 | 0.88 | 0.91 | 0.93 | 0.79 | 0.82 | 0.90 | 0.92 | 0.93 | 0.94 | 0.95 | 0.96 |
| (20,20,10,5) | 0.85 | 0.86 | 0.91 | 0.91 | 0.79 | 0.81 | 0.89 | 0.92 | 0.93 | 0.92 | 0.95 | 0.95 |
| Target → | (20,10) | | | | (20,10,5) | | | | (20,20,10,5) | | | |
| $\mathbb{D}_{\text{uni}}$ | n=100 | | n=200 | | n=100 | | n=200 | | n=100 | | n=200 | |
| Surrogate ↓ | Base. | CCA | Base. | CCA | Base. | CCA | Base. | CCA | Base. | CCA | Base. | CCA |
| (20,10) | 0.71 | 0.81 | 0.75 | 0.87 | 0.78 | 0.84 | 0.79 | 0.87 | 0.84 | 0.88 | 0.85 | 0.91 |
| (20,10,5) | 0.71 | 0.78 | 0.74 | 0.83 | 0.77 | 0.82 | 0.78 | 0.85 | 0.82 | 0.88 | 0.84 | 0.90 |
| (20,20,10,5) | 0.71 | 0.75 | 0.74 | 0.81 | 0.77 | 0.81 | 0.78 | 0.84 | 0.82 | 0.86 | 0.84 | 0.90 |

Table 4: Fidelity over $\mathbb{D}_{\text{test}}$ and $\mathbb{D}_{\text{uni}}$ for COMPAS dataset

| Target → | (20,10) | | | | (20,10,5) | | | | (20,20,10,5) | | | |
|---|---|---|---|---|---|---|---|---|---|---|---|---|
| $\mathbb{D}_{\text{test}}$ | n=100 | | n=200 | | n=100 | | n=200 | | n=100 | | n=200 | |
| Surrogate ↓ | Base. | CCA | Base. | CCA | Base. | CCA | Base. | CCA | Base. | CCA | Base. | CCA |
| (20,10) | 0.93 | 0.96 | 0.94 | 0.97 | 0.92 | 0.94 | 0.94 | 0.96 | 0.94 | 0.96 | 0.95 | 0.97 |
| (20,10,5) | 0.92 | 0.95 | 0.94 | 0.97 | 0.92 | 0.93 | 0.95 | 0.95 | 0.94 | 0.96 | 0.95 | 0.97 |
| (20,20,10,5) | 0.92 | 0.95 | 0.92 | 0.97 | 0.84 | 0.91 | 0.89 | 0.94 | 0.92 | 0.94 | 0.94 | 0.96 |
| Target → | (20,10) | | | | (20,10,5) | | | | (20,20,10,5) | | | |
| $\mathbb{D}_{\text{uni}}$ | n=100 | | n=200 | | n=100 | | n=200 | | n=100 | | n=200 | |
| Surrogate ↓ | Base. | CCA | Base. | CCA | Base. | CCA | Base. | CCA | Base. | CCA | Base. | CCA |
| (20,10) | 0.94 | 0.95 | 0.94 | 0.96 | 0.95 | 0.95 | 0.95 | 0.96 | 0.96 | 0.95 | 0.96 | 0.96 |
| (20,10,5) | 0.93 | 0.95 | 0.94 | 0.95 | 0.94 | 0.92 | 0.95 | 0.92 | 0.95 | 0.96 | 0.96 | 0.96 |
| (20,20,10,5) | 0.93 | 0.94 | 0.94 | 0.95 | 0.94 | 0.85 | 0.94 | 0.90 | 0.95 | 0.92 | 0.95 | 0.94 |

Table 5: Fidelity over $\mathbb{D}_{\text{test}}$ and $\mathbb{D}_{\text{uni}}$ for HELOC dataset

| Target $\rightarrow$ | (20,10) | | | | (20,10,5) | | | | (20,20,10,5) | | | |
|---|---|---|---|---|---|---|---|---|---|---|---|---|
| $\mathbb{D}_{\text{test}}$ | n=100 | | n=200 | | n=100 | | n=200 | | n=100 | | n=200 | |
| Surrogate $\downarrow$ | Base. | CCA | Base. | CCA | Base. | CCA | Base. | CCA | Base. | CCA | Base. | CCA |
| (20,10) | 0.90 | 0.94 | 0.91 | 0.95 | 0.90 | 0.94 | 0.92 | 0.95 | 0.98 | 0.99 | 0.98 | 0.99 |
| (20,10,5) | 0.88 | 0.92 | 0.92 | 0.95 | 0.89 | 0.92 | 0.92 | 0.95 | 0.98 | 0.98 | 0.98 | 0.99 |
| (20,20,10,5) | 0.87 | 0.93 | 0.91 | 0.93 | 0.87 | 0.89 | 0.91 | 0.94 | 0.98 | 0.98 | 0.98 | 0.98 |
| Target $\rightarrow$ | (20,10) | | | | (20,10,5) | | | | (20,20,10,5) | | | |
| $\mathbb{D}_{\text{uni}}$ | n=100 | | n=200 | | n=100 | | n=200 | | n=100 | | n=200 | |
| Surrogate $\downarrow$ | Base. | CCA | Base. | CCA | Base. | CCA | Base. | CCA | Base. | CCA | Base. | CCA |
| (20,10) | 0.92 | 0.92 | 0.94 | 0.95 | 0.91 | 0.91 | 0.94 | 0.95 | 0.98 | 0.98 | 0.99 | 0.99 |
| (20,10,5) | 0.91 | 0.90 | 0.94 | 0.93 | 0.91 | 0.89 | 0.93 | 0.94 | 0.97 | 0.97 | 0.98 | 0.99 |
| (20,20,10,5) | 0.91 | 0.91 | 0.93 | 0.94 | 0.91 | 0.87 | 0.93 | 0.92 | 0.97 | 0.97 | 0.98 | 0.98 |

### B.2.6 Studying alternate counterfactual generating method

Counterfactuals can be generated such that they satisfy additional desirable properties such as actionability, sparsity and closeness to the data manifold, other than the proximity to the original instance. In this experiment, we observe how counterfactuals with above properties affect the attack performance. HELOC is used as the dataset. Target model has the architecture (20, 30, 10) and the architecture of the surrogate model is (10, 20).

To generate actionable counterfactuals, we use Diverse Counterfactual Explanations (DiCE) by Mothilal et al. [2020] with the first four features, i.e., "estimate_of_risk", "months_since_last_trade", "average_duration_of_resolution", and "number_of_satisfactory_trades" kept unchanged. The diversity factor of DiCE generator is set to 1 in order to obtain only a single counterfactual for each query. Sparse counterfactuals are obtained by the same MCCF generator used in other experiments, but now with $L_1$ norm as the cost function $c(\boldsymbol{x}, \boldsymbol{w})$. Counterfactuals from the data manifold (i.e., realistic counterfactuals, denoted by 1-NN) are generated using a 1-Nearest-Neighbor algorithm. We use ROAR [Upadhyay et al., 2021] and C-CHVAE [Pawelczyk et al., 2020] to generate robust counterfactuals. Table 2 summarizes the performance of the attack. Fig. 8 shows the distribution of the counterfactuals generated using each method w.r.t. the decision boundary of the target model. We observe that the sparse, realistic, and robust counterfactuals have a tendency to lie farther away from the decision boundary, within the favorable region, when compared to the closest counterfactuals under $L_2$ norm.

### B.2.7 Comparison with DualCFX Wang et al. [2022]

Wang et al. [2022] is one of the few pioneering works studying the effects of counterfactuals on model extraction, which proposes the interesting idea of using counterfactuals of counterfactuals to mitigate the decision boundary shift. This requires the API to provide counterfactuals for queries originating from both sides of the decision boundary. However, the primary focus of our work is on the one-sided scenario where an institution might be giving counterfactuals only to the rejected applicants to help them get accepted, but not to the accepted ones. Hence, a fair comparison cannot be achieved between CCA and the strategy proposed in Wang et al. [2022] in the scenario where only one-sided counterfactuals are available.

Therefore, in the two-sided scenario, we compare the performance of CCA with the DualCFX strategy proposed in Wang et al. [2022] under two settings:

1. only one sided counterfactuals are available for CCA (named CCA1)
2. CCA has all the data that DualCFX has (named CCA2)

We also include another baseline (following Aïvodji et al. [2020]) for the two-sided scenario where the models are trained only on query instances and counterfactuals, but not the counterfactuals of the counterfactuals. Results are presented in Table 6. Note that even for the same number of initial

query instances, the total number of actual training instances change with the strategy being used (CCA1 < Baseline < DualCFX = CCA2 – e.g.: queries+CFs for the baseline but queries+CFs+CCFs for DualCFX).

Table 6: Comparison with DualCFX. Legend: Base.=Baseline model based on [Aïvodji et al., 2020], Dual=DualCFX, CCA1=CCA with one-sided counterfactuals, CCA2=CCA with counterfactuals from both sides.

| Dataset | Query size | Architecture known (model 0) | | | | | | | |
| | | $\mathbb{D}_{\text{test}}$ | | | | $\mathbb{D}_{\text{uni}}$ | | | |
| | | Base. | Dual. | CCA1 | CCA2 | Base. | Dual. | CCA1 | CCA2 |
| DCCC | n=100 | 0.95 | 0.99 | 0.94 | 0.99 | 0.90 | 0.95 | 0.92 | 0.97 |
| | n=200 | 0.96 | 0.99 | 0.98 | 0.99 | 0.90 | 0.96 | 0.95 | 0.98 |
| HELOC | n=100 | 0.94 | 0.97 | 0.90 | 0.98 | 0.91 | 0.98 | 0.84 | 0.98 |
| | n=200 | 0.96 | 0.98 | 0.92 | 0.98 | 0.93 | 0.98 | 0.89 | 0.99 |

| Dataset | Query size | Architecture unknown (model 1) | | | | | | | |
| | | $\mathbb{D}_{\text{test}}$ | | | | $\mathbb{D}_{\text{uni}}$ | | | |
| | | Base. | Dual. | CCA1 | CCA2 | Base. | Dual. | CCA1 | CCA2 |
| DCCC | n=100 | 0.92 | 0.98 | 0.93 | 0.98 | 0.88 | 0.92 | 0.89 | 0.93 |
| | n=200 | 0.96 | 0.99 | 0.96 | 0.99 | 0.89 | 0.94 | 0.94 | 0.96 |
| HELOC | n=100 | 0.92 | 0.91 | 0.90 | 0.96 | 0.88 | 0.92 | 0.84 | 0.96 |
| | n=200 | 0.95 | 0.92 | 0.91 | 0.97 | 0.93 | 0.94 | 0.88 | 0.97 |

### B.2.8  Studying other machine learning models

We explore the effectiveness of the proposed attack when the target model is no longer a neural network classifier. The surrogate models are still neural networks with the architectures (20, 10) for model 0 and (20, 10, 5) for model 1. A random forest classifier with 100 estimators and a linear regression classifier, trained on Adult Income dataset are used as the targets. Ensemble size $S$ used is 20. Results are shown in Fig. 16, where the proposed attack performs better or similar to the baseline attack.

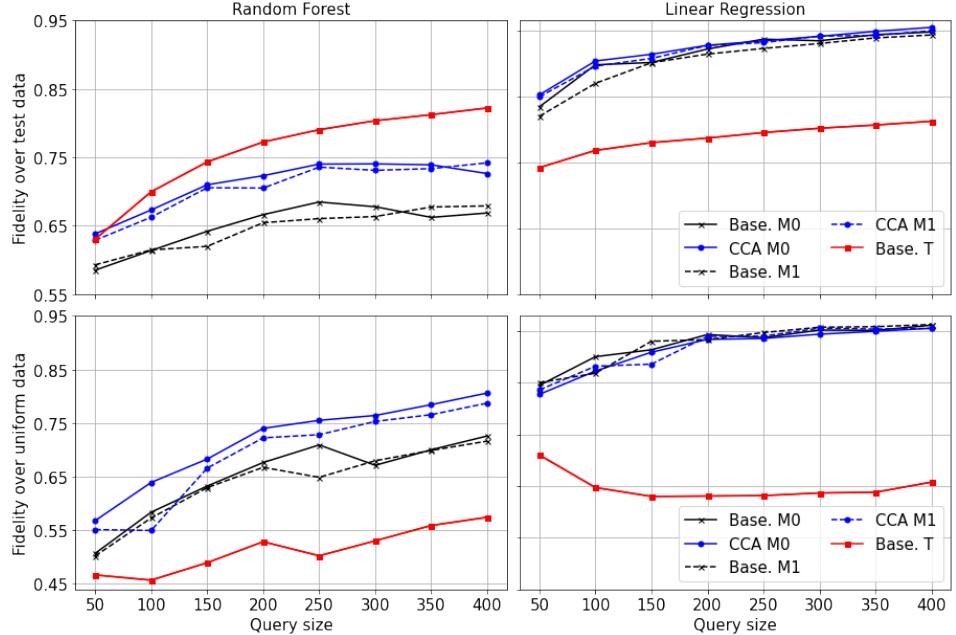

Figure 16: Performance of the attack when the target model is not a neural network. Surrogates M0 and M1 are neural networks with the architectures (20,10) and (20,10,5) respectively. Baseline T is a surrogate model from the same class as the target model.

### B.2.9 Studying effect of unbalanced $\mathbb{D}_{\text{attack}}$

In all the other experiments, the attack dataset $\mathbb{D}_{\text{attack}}$ used by the adversary is sampled from a class-wise balanced dataset. In this experiment we explore the effect of querying using an unbalanced $\mathbb{D}_{\text{attack}}$. Model architectures used are (20, 10) for the target model and surrogate model 0, and (20, 10, 5) for surrogate model 1. While the training set of the teacher and the test set of both the teacher and the surrogates were kept constant, the proportion of the samples in the attack set $\mathbb{D}_{\text{attack}}$ was changed. In the first case, examples from class $y = 1$ were dominant (80%) and in the second case, the majority of the examples were from class $y = 0$ (80%). The results are shown in Fig. 17.

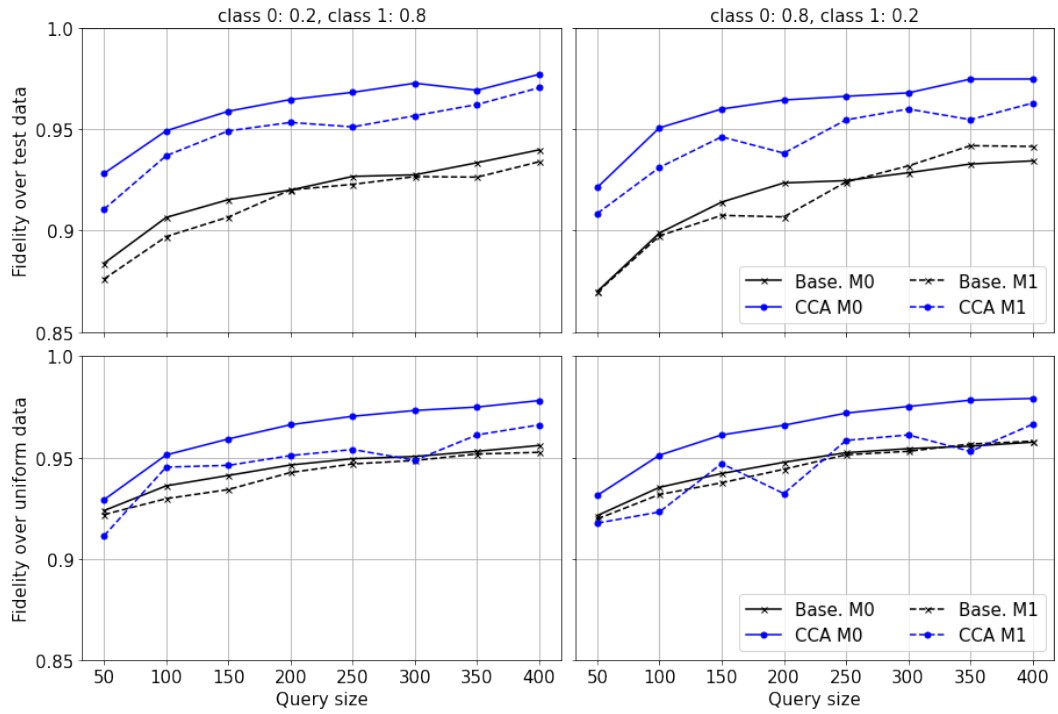

Figure 17: Results corresponding to the HELOC dataset with queries sampled from biased versions of the dataset (i.e., a biased $\mathbb{D}_{\text{attack}}$). The version on the left uses a $\mathbb{D}_{\text{attack}}$ with 20% and 80% examples from classes $y = 0$ and $y = 1$, respectively. The version on the right was obtained with a $\mathbb{D}_{\text{attack}}$ comprising of 80% and 20% examples from classes $y = 0$ and $y = 1$, respectively.

### B.2.10 Studying alternate loss functions

We explore using binary cross-entropy loss function directly with labels 0, 1 and 0.5 in place of the proposed loss. Precisely, the surrogate loss is now defined as

$$L(\tilde{m}, y) = -y(\boldsymbol{x}) \log\left(\tilde{m}(\boldsymbol{x})\right) - (1 - y(\boldsymbol{x})) \log\left(1 - \tilde{m}(\boldsymbol{x})\right) \tag{32}$$

which is symmetric around 0.5 for $y(\boldsymbol{x}) = 0.5$. Two surrogate models are observed, with architectures (20, 10) for model 0 and (20, 10, 5) for model 1. The target model's architecture is similar to that of model 0. The ensemble size is $S = 20$.

The results (in Fig. 18) indicate that the binary cross-entropy loss performs worse than the proposed loss. The reason might be the following: As the binary cross-entropy loss is symmetric around 0.5 for counterfactuals, it penalizes the counterfactuals that are farther inside the favorable region. This in turn pulls the surrogate decision boundary towards the favorable region more than necessary, causing a decision boundary shift.

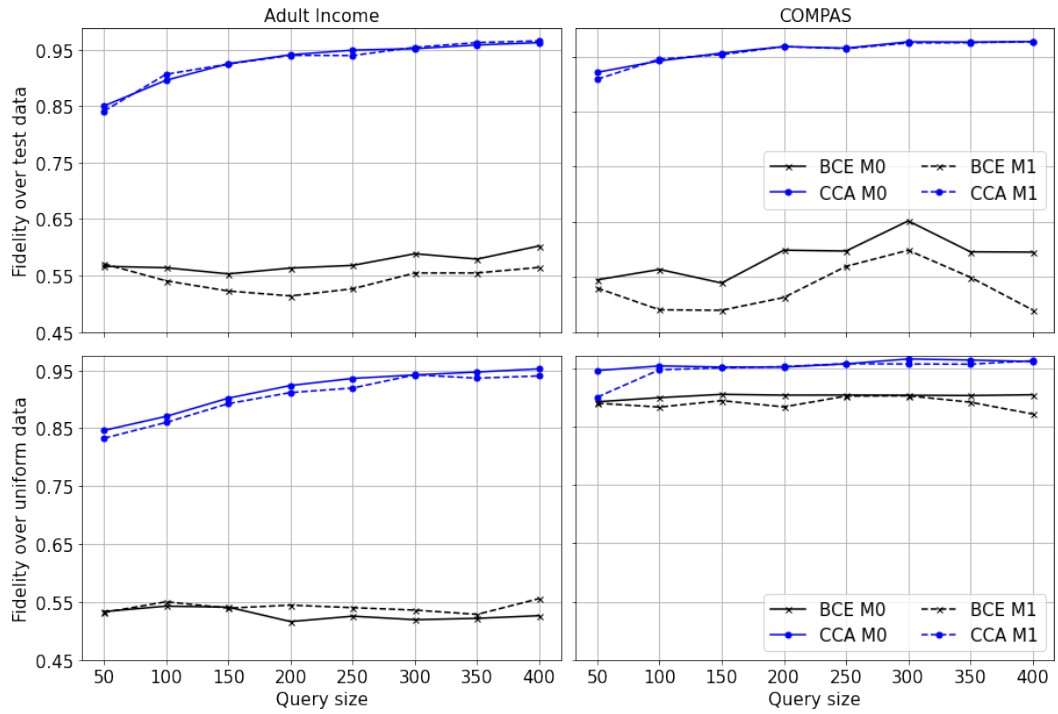

Figure 18: Performance of binary cross-entropy loss with labels 0, 0.5 and 1. Black lines corresponding to binary cross entropy (BCE) loss and blue lines depict the performance of the CCA loss.

## B.3 Experiments for verifying Theorem 3.2

This experiment includes approximating a spherical decision boundary in the first quadrant of a $d-$dimensional space. The decision boundary is a portion of a sphere with radius 1 and the origin at $(1, 1, \ldots, 1)$. The input space is assumed to be normalized, and hence, restricted to the unit hypercube. See Section 3.1 for a description of the attack strategy. Fig. 19 presents a visualization of the experiment in the case where the dimensionality $d = 2$. Fig. 20 presents a comparison of theoretical and empirical query complexities for higher dimensions. Experiments agree with the theoretical upper-bound.

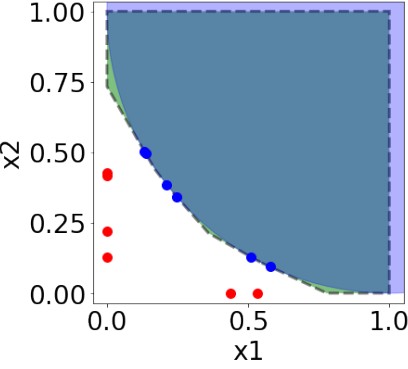

Figure 19: Synthetic attack for verifying Theorem 3.2 in the 2-dimensional case. Red dots represent queries and blue dots are the corresponding closest counterfactuals. Dashed lines indicate the boundary of the polytope approximation.

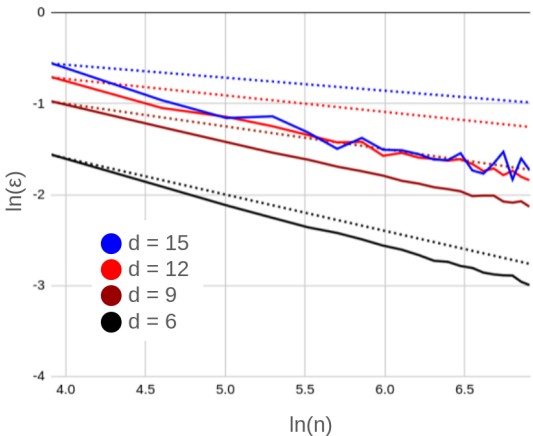

Figure 20: Verifying Theorem 3.2: Dotted and solid lines indicate the theoretical and empirical rates of convergence.

