# OpenReview forum: "Model Reconstruction Using Counterfactual Explanations: A Perspective From Polytope Theory"
_NeurIPS.cc/2024/Conference — NeurIPS 2024 poster_

### Official Review · Reviewer_J5Uq · 2024-06-16

**Soundness:** 3
**Presentation:** 3
**Contribution:** 3
**Rating:** 6
**Confidence:** 3

**Summary:**

This paper presents a novel and valuable theoretical analysis leveraging polytope theory to provide novel insights into using counterfactual explanations for model reconstruction and significant contribution to the field of model reconstruction/interpretability of black box models.  The key contributions are:
Providing a geometric interpretation of how counterfactuals relate to the decision boundary using polytope geometry.
Deriving bounds on the expected approximation error of the reconstructed model as a function of the number of counterfactual queries .
Extending the analysis to handle cases where only approximate closest counterfactuals are available using Lipschitz continuity.
Proposing the Counterfactual Clamping Attack (CCA) which treats counterfactuals differently in the loss function to prevent decision boundary shift during training of the surrogate model.
Demonstrating CCA's improved performance over baselines on multiple real-world datasets, including with one-sided counterfactuals.

**Strengths:**

Providing novel theoretical insights into the relationship between counterfactuals and the decision boundary geometry using polytope theory.
Deriving mathematical bounds relating approximation error to the number of counterfactual queries.
Proposing CCA which mitigates decision boundary shift, a key issue with prior counterfactual-based model extraction approaches.
Extensive empirical evaluation validating CCA's effectiveness across datasets.

**Weaknesses:**

Limited discussion on the applicability of CCA to model families beyond neural networks like tree based models XGBoost, Random Forest which are most commonly used for problems discussed like loan decisions. The loss function and counterfactual generation process may need adaptations.
Lack of analysis on the computational complexity and scalability of CCA, especially for high-dimensional data or large datasets. Generating closest counterfactuals can be computationally expensive.
No exploration of the effect of counterfactual quality aspects like sparsity, actionability or realness on CCA's performance. The paper does not analyze how enforcing quality constraints on counterfactuals impacts CCA's ability to accurately reconstruct the target model's decision boundary.

**Questions:**

1. Have you explored the application of CCA to other models like tree ensembles, XGBoost or others which might have non continuous / non differentiable decision boundaries?
2. Have you explored into the computational complexity and memory requirements of CCA, especially when scaling to high-dimensional data or large datasets for example in comparison to other standard model reconstruction methods for practical applications of the approach?
3. Any exploration of sensitivity of CCA's performance to the quality (e.g., sparsity, realistic) of the counterfactual instances used?

**Limitations:**

The theoretical analysis is limited to neural network architectures and assumes convex/concave decision boundaries. The authors should discuss the applicability to other model families like tree ensembles.
While bounds on approximation error are derived, there is limited analysis of the computational complexity and scalability of the proposed Counterfactual Clamping Attack (CCA) to high-dimensional data or large datasets.
The paper does not explore the effect of counterfactual quality criteria like sparsity and similarity on CCA's model extraction performance, which is important for real-world applications.
The empirical evaluation is focused on classification tasks. More analysis on regression problems or tasks with complex decision boundaries is needed.

---

> ### Author Rebuttal · Authors · 2024-08-07
>
> We thank the reviewer for reading the paper and appreciate their detailed review.
>
> **_CCA for other machine learning models:_** The proposed CCA algorithm, being implemented through a modified loss function, in its current form is limited to neural networks. However, the initial theoretical development, specifically, Theorem 3.2 and Corollary 3.11 are applicable to models with a convex decision region or models that are Lipschitz continuous, respectively. Extending Algorithm 1 to other models, as suggested by the reviewer, is an important path for exploration in the future.
>
> **_Computational complexity:_** In comparison to existing strategies of **model extraction specifically using counterfactuals** [Aïvodji et al., Wang et al.], our model training process in Algorithm 1 is the usual, except for the two factors: the loss function and the training dataset. Our loss function does not have any significant impact on the computational complexity or the memory requirement, since it involves typical mathematical operations such as addition and multiplication.
>
> On the other hand, the training set includes counterfactual explanations and regular datapoints. We note that **computing counterfactual explanations can be computationally intensive,** particularly when the data is high-dimensional, and also varies based on the generation method chosen. We assume this increased computational burden is on the API’s side. Therefore, the reconstruction strategy is not directly affected by the computational complexity or memory requirements of computing the counterfactuals.
>
> Though, we acknowledge that it would indeed be **an interesting discussion** if considering the overall computational complexity of both parties including both model training and the API’s counterfactual generation complexity. [Aïvodji et al.] shows that model extraction using counterfactuals has better performance using fewer queries than traditional methods which do not use counterfactuals. DualCFX [Wang et al.] further improves performance at the expense of requiring the counterfactual of counterfactuals. Our work eliminates the requirement of the counterfactuals of counterfactuals by specifically leveraging the fact that counterfactuals are close to the decision boundary. **The benefit of model extraction specifically using counterfactuals does come at the additional computational cost of generating counterfactuals in the first place.** The exact cost depends on the strategy being used.
>
> **_Sensitivity to the quality of counterfactuals:_** We considered the following counterfactual generating methods in the manuscript, including MCCF with L1 and L2 norms, DiCE actionable counterfactuals, 1-Nearest-Neighbor counterfactuals (counterfactuals from the data manifold) and DiCE counterfactuals with varying levels of sparsity (Section D.2.6 Table 5 and Fig. 16, 17). Now, we also include ROAR [Upadhyay et al.] and C-CHVAE [Pawelczyk et al.]  and present the consolidated results in Rebuttal PDF Table 7. We have also included histograms of the prediction probabilities for counterfactuals generated by different methods.
>
> We would like to highlight the fact that our strategy does not take into account any specifics of the generating method. Instead, what affects the performance is the distribution of the counterfactuals around the decision boundary. Histograms in the Rebuttal PDF Fig. 23 provide some insights on how the counterfactuals generated using different methods are distributed. Firstly, observe that our strategy CCA does not require closest counterfactuals but is able to reconstruct models quite faithfully for many of the other counterfactual generating methods even if they are not the closest.
>
> Additionally, we also observe that the robust counterfactuals generated using the ROAR method have relatively higher prediction probabilities from the target model. As we may observe from these histograms, when the counterfactual distribution is concentrated around higher prediction probabilities (e.g.: in case of ROAR or sparse DiCE with posthoc_sparsity_param=0.2 – see manuscript section D.2.6. Fig. 16, 17), the advantage of CCA over the baseline diminishes.
>
> We will include a detailed discussion on these observations in the paper.
>
> [Wang et al.] Wang, Y., Qian, H., & Miao, C. (2022, June). Dualcf: Efficient model extraction attack from counterfactual explanations. In Proceedings of the 2022 ACM Conference on Fairness, Accountability, and Transparency (pp. 1318-1329).
>
> [Aïvodji et al.] Aïvodji, U., Bolot, A., & Gambs, S. (2020). Model extraction from counterfactual explanations. arXiv preprint arXiv:2009.01884.
>
> [Upadhyay et al.] Upadhyay, S., Joshi, S., & Lakkaraju, H. (2021). Towards robust and reliable algorithmic recourse. Advances in Neural Information Processing Systems, 34, 16926-16937.
>
> [Pawelczyk et al.] Pawelczyk, M., Broelemann, K., & Kasneci, G. (2020, April). Learning model-agnostic counterfactual explanations for tabular data. In Proceedings of the web conference 2020 (pp. 3126-3132).

---

> > ### Comment · Reviewer_J5Uq · 2024-08-12
> > **Author Response acknowledged**
> >
> > I have read the authors response.

---

> > > ### Author Response · Authors · 2024-08-13
> > >
> > > Thank you very much for raising these important questions and the detailed review. We are ready to address any other questions the reviewer might have.

---

### Official Review · Reviewer_z1A7 · 2024-07-11

**Soundness:** 2
**Presentation:** 3
**Contribution:** 2
**Rating:** 5
**Confidence:** 4

**Summary:**

The paper proposed a model reconstruction methodology by using one-sided counterfactual explanations. After generating counterfactuals (assumed to be the closest counterfactual to the observation), the authors reconstruct the original model using a piece-wise linear approximation (with a bunch of hyperplanes). The key idea is that such hyperplanes can be identified by the pair of observation and its closest counterfactual, of which the the joining-line is straightforwardly perpendicular to the model's decision boundary. The paper discussed convex and non-convex decision regions, and demonstrated that the general non-convex case is challenging for being approximated with finite number of counterfactual explanations. Nevertheless, the authors showed that given sufficiently many counterfactual explanations (queries), a ReLU neural network can still be reconstructed. The experiment results demonstrated improvement over one literature (served as baseline), for both the known architecture and unknown architecture of the ReLU network.

The paper is well-written and organized. The idea, to my knowledge, is novel. The paper goes into good depth for analysis, showing interesting insights.

**Strengths:**

The idea of using counterfactual explanations together with piece-wise linear approximation for model reconstruction, to my knowledge, is novel. The quality of writing and presentation is fairly clear.

**Weaknesses:**

The key limitation of this paper is that all the theories hold if the counterfactuals are the closest ones to the observations, which is difficult to be guaranteed utilizing counterfactual explanations algorithms. One way to deal with this is to consider in the derived theory some tolerance on the counterfactual's quality. Namely, if an algorithm is able to find counterfactuals within a given radius epsilon of the closest counterfactual, then how does this affect model construction.

The connection between the proposed theory and the effectiveness of the authors model construction methodology is not stated. Namely, algorithm 1 could stand alone without any of the derived theories.

Theorem 3.6 relies on the assumption that each cell only contains completely linear part of the model's decision boundary, with one closest counterfactual in each cell. In order to make the first part (that only linear part is contained in a cell) holds for most of the cells, one has to make the edge of the cell (varepsilon) really small, which requires a massive amount of queries to be done. Hence, the assumption that is required to make this Theorem work, is too strong.

**Questions:**

How does the "polytope theory" help the model reconstruction algorithm design? Namely, the foundation (Lemma 3.1), the fidelity expectation for the convex case (Theorem 3.2), and the reconstruction probability lower bound (Theorem 3.6) are proposed, but how could we use them to better reconstruct the model than existing counterfactual explanations based reconstruction methodologies?

The results in Table 1 have large confidence intervals. For example, in the column Architecture unknown (model 1), D_{uni}, and row DCCC, the baseline is 95+-2.2 and the proposed CCA is 95+-11.8 etc. Does this imply that the proposed method is not very stable?

What if we assume that sufficient many queries can be made and then increase the number of queries from 400 to 4000? Will the performance of CCA go close to the baseline?

**Limitations:**

Yes, the authors have stated clearly some limitations, and foresee potential non-positive impact.

---

> ### Author Rebuttal · Authors · 2024-08-07
>
> We thank the reviewer for taking out the time to review this paper, and appreciate their acknowledgement of the novelty of our work.
>
> **_On significance and limitations of theoretical results:_** Deriving theoretical bounds for model reconstruction using counterfactuals under general non-convex decision boundaries is a challenging problem. Our work proposes a novel non-trivial approach towards solving this problem through the lens of polytopes with assumptions for mathematical tractability. We will further elaborate on these assumptions in our limitations.
>
> 1. **Closest counterfactuals:** We agree that assuming the availability of the closest counterfactuals in Theorems 3.2 and 3.6 limits their direct applicability into scenarios where the counterfactuals are reasonably close but not exactly the closest one. However, Theorems 3.2 and 3.6 still provide valuable insights into how counterfactuals might facilitate model reconstruction. E.g.: the empirical mean approximation error appears to be of $o(n^{-2/(d-1)})$ which is the theoretical rate of decay derived by assuming closest counterfactuals and convex decision boundaries (as discussed in Section D.2.3 of the manuscript).
>
> 2. **Additional insights:** Moreover, Theorem 3.6 suggests that target models with simpler decision boundaries are easier to extract, and the Corollary 3.8 specifically shows why model reconstruction is mathematically easier when two-sided counterfactuals are available. Corollary 3.8 is quite relevant in the context of existing literature on model extraction which typically assume two-sided counterfactuals are available. For instance, an institution might give counterfactuals to rejected applicants but not necessarily to the accepted ones, leading to the one-sided counterfactual scenario.
>
> 3. **Beyond closest counterfactuals:** In Section 3.3, we no longer assume the counterfactuals to be the closest, but they lie on the decision boundary, and provide guarantees under local Lipschitz assumptions (Theorem 3.10 and Corollary 3.11). We greatly appreciate the insightful suggestion to consider **a tolerance ball**, which can be included directly into Corollary 3.11. We will also include this suggestion in future works in conjunction with other relaxations such as probabilistically-Lipschitz assumptions [Khan et al.]. Algorithm 1 is motivated from the need to clamp the counterfactuals to the decision boundary as per our analysis in Theorem 3.10 and Corollary 3.11 (which will have further relaxations now using the tolerance ball).
>
> 4. **Grid size:** Approximating a non-convex polytope becomes challenging due to facts such as it being impossible to be simply expressed as an intersection of half-planes leading us to explore ReLu networks whose decision boundaries are polytopes allowing for some mathematical traceability if the boundary is locally linear over some cells of a grid. We agree that the edge-cells (cells in which two or more high-dimensional facets are present) violate the assumption and hence, the grid size needs to be reduced (though such edge cells might be fewer in comparison to the entire set of boundary cells).
>
> Nonetheless, we believe that our approach is a non-trivial approach towards solving this challenging problem. **The number of linear regions of a ReLU network in practice has been observed to be far less than the theoretically achievable maximum** [Hanin and Rolnick]. Moreover, [Jordan et al.] suggests that the decision regions can be represented as polyhedral complexes, which are a specific type of unions of convex polytopes. This may further reduce the number of high-dimensional edges that would occur in practice, as opposed to the worst-case scenario. Therefore, the required size of the grid might actually depend largely on the complexity of the classification problem.
>
> **_Variability of results in Table 1:_** The results in Table 1 have been averaged after training 100 different target models, generating queries and counterfactuals for them multiple times, and then surrogate models for each case. As pointed out by the reviewer, the standard deviation is a bit high for one setup, but for most others, it is fairly reasonable and comparable to the existing methods of model extraction using counterfactuals.
>
> **_Significantly higher number of queries:_** For a very high number of queries, the performance depends on how the positive and negative queries are actually distributed. Sometimes, we observe that good fidelity is achieved for both CCA and baseline possibly because the positive and negative queries dominate a lot more over the counterfactuals. Though, in a few cases, the fidelity of the baseline does not keep increasing with the number of queries and saturates because it treats the counterfactuals as points with label 1, and hence suffers from decision boundary shift issue (in this situation, ignoring the counterfactuals entirely or using CCA might be helpful than treating them as label 1 instances).
>
>
> [Khan et al.] Khan, Z. Q., Hill, D., Masoomi, A., Bone, J. T., & Dy, J. (2024, April). Analyzing Explainer Robustness via Probabilistic Lipschitzness of Prediction Functions. In International Conference on Artificial Intelligence and Statistics (pp. 1378-1386). PMLR.
>
> [Hanin and Rolnick] Hanin, B., & Rolnick, D. (2019, May). Complexity of linear regions in deep networks. In International Conference on Machine Learning (pp. 2596-2604). PMLR.
>
> [Jordan et al.] Jordan, M., Lewis, J., & Dimakis, A. G. (2019). Provable certificates for adversarial examples: Fitting a ball in the union of polytopes. Advances in neural information processing systems, 32.

---

> > ### Comment · Reviewer_z1A7 · 2024-08-08
> >
> > Thank you for your response.
> > > The number of linear regions of a ReLU network in practice has been observed to be far less than the theoretically achievable maximum [Hanin and Rolnick]
> >
> > Do you want to say that the "corners" in Figure 5 would be quite many such that in reality we need many grids to properly use Theorem 3.6, or the opposite? It reads a bit confusion to me.
> >
> > > Moreover, [Jordan et al.] suggests that the decision regions can be represented as polyhedral complexes, which are a specific type of unions of convex polytopes. This may further reduce the number of high-dimensional edges that would occur in practice, as opposed to the worst-case scenario. Therefore, the required size of the grid might actually depend largely on the complexity of the classification problem.
> >
> > I believe it worths a discussion somewhere in the paper, especially for how the size of the grid could be influenced in practice. One of the major concerns of the proposed method, from my point of view, lies on the two strong assumptions (though they are not assumed simultaneously):
> > 1. "Closeness" for convex case
> > 2. "Grids num" required for non-convex case
> >
> > Still, I'm not fully convinced by your response (I meant, I believe in sentences your argument, but they seem not persuasive as rebuttal for why these two assumptions are not too strong.)
> >
> > > Algorithm 1 is motivated from the need to clamp the counterfactuals to the decision boundary as per our analysis in Theorem 3.10 and Corollary 3.11 (which will have further relaxations now using the tolerance ball).
> > And yet, I don't think this answers my question fully. My original question is how does the "polytope theory" help the model reconstruction algorithm design, since "A perspective from polytope theory" is part of your title? Namely, how do the bounds you have derived, contribute to Algorithm 1? To me, Algorithm 1 seems quite independent to other parts of the paper and can stand alone very well.
> >
> > But I believe there are shining and inspiring points in this paper, even though with above weaknesses. So, I would like to keep my score.

---

> > > ### Author Response · Authors · 2024-08-12
> > > **Reply by Authors**
> > >
> > > We appreciate that the reviewer finds that “there are shining and inspiring points in this paper.” We will definitely include a detailed discussion on the assumptions that we have to make for analytical tractability and the limitations pointed out by the Reviewer.
> > >
> > > **_Clarifications on the grid-size ($\epsilon$) assumption:_** In general, approximating a non-convex decision boundary becomes challenging since it is impossible to express it as an intersection of half-planes. However, for a ReLU network, the problem becomes analytically tractable because the input domain can be partitioned into **q** regions such that the model is linear in each region (leading to polytope decision boundaries with at most **q** linear pieces) [Chen et al., Hanin and Rolnick]. In fact, if the number of such partitioned regions is q and **one assumes that the adversary knows each of these partitions**, we can derive another result similar to our Theorem 3.6 by constructing only q inverse counterfactual regions and the probability will depend on only q (and not the grid-size).
> > >
> > > $$\mathbb{P}[\text{Reconstruction}] \geq 1-q(1-v^*)^n$$
> > > where $v^* = \min_i v_i$ with $v_i$ being the volume of the $i^\text{th}$ inverse counterfactual region ($i=1,2,..,q$).
> > >
> > >
> > > However, instead of assuming that the adversary exactly knows the partitions of the ReLU network they are trying to reconstruct (which we feel would be a much stronger assumption for our problem), we assume there is a uniform grid such that the ReLU network’s decision boundary is linear across each small grid-cell that it passes through.
> > >
> > > We agree that the grid-size would need to be reduced in order for this to hold: otherwise, there would be more “edge” cells where two or more facets intersect (e.g., cells containing the corners in 2D). Nonetheless, it is still a weaker assumption than assuming the adversary exactly knows the partitions of the ReLU network.
> > >
> > > Furthermore, the grid-size would still be determined by the complexity of the ReLU network’s decision boundary (essentially goes back to the number of original partitions q). What we wanted to highlight is that this number of partitions for a ReLU network observed in practice is far less than the theoretically possible maximum [Hanin and Rolnick], holding promise that the grid size might also not need to be too small.
> > >
> > > Moreover, [Jordan et al.] suggests that the decision regions can be represented as polyhedral complexes, which are a specific type of unions of convex polytopes. This may further reduce the number of high-dimensional edges that would occur in practice, as opposed to the worst-case scenario, and consequently holds promise that the number of such “edge” cells (cells containing corners in 2D) would be fewer. Therefore, the required size of the grid might actually depend largely on the complexity of the classification problem.
> > >
> > > Relaxing the said assumptions is the main goal of our future work. In particular, we will study the possibility of allowing the adversary to know alternate partitions such that the ReLU network is linear over each region.
> > >
> > >
> > > **_On Algorithm 1:_** Algorithm 1 can stand on its own, without the basis of theorems 3.2 and 3.6 (Theorem 3.10 and Corollary 3.11 provide the main intuition for Algorithm 1). However we see theorems 3.2 and 3.6 as important parts of the journey where we start from a more constrained but mathematically-tractable setting and move to a less constrained but not-so-mathematically-tractable setting.
> > >
> > > Thank you very much for the detailed review and the valuable insights.
> > >
> > >
> > > [Chen et al.] Chen, K. L., Garudadri, H., & Rao, B. D. (2022). Improved bounds on neural complexity for representing piecewise linear functions. Advances in Neural Information Processing Systems, 35, 7167-7180.
> > >
> > > [Hanin and Rolnick] Hanin, B., & Rolnick, D. (2019, May). Complexity of linear regions in deep networks. In International Conference on Machine Learning (pp. 2596-2604). PMLR.
> > >
> > > [Jordan et al.] Jordan, M., Lewis, J., & Dimakis, A. G. (2019). Provable certificates for adversarial examples: Fitting a ball in the union of polytopes. Advances in neural information processing systems, 32.

---

### Official Review · Reviewer_Bnbn · 2024-07-11

**Soundness:** 3
**Presentation:** 3
**Contribution:** 3
**Rating:** 6
**Confidence:** 3

**Summary:**

The paper presents an approach to mitigate the decision boundary shift problem of counterfactual-based model reconstruction algorithms. They leverage the fact that counterfactuals differ from ordinary instances since they exist relatively close to the decision boundary, to derive a novel loss function for model reconstruction that provides special treatment to counterfactuals. The theoretical foundations of this work are based on polytope theory, using which the authors derive a relationships between the error in model reconstruction and the number of required counterfactual samples. Experimental results show that the proposed approach surpasses the baseline on a number of model reconstruction benchmarks.

**Strengths:**

1. The paper is very clearly written. The authors set a concise and understandable stage for the problem of model reconstruction using counterfactual explanations, and explaining the limitation of decision boundary shift that the existing models face due to treating counterfactuals same as ordinary samples. They zero-in on their approach motivated by a sound theoretical basis which establishes a relationship between the error in model reconstruction and number of required counterfactual examples using polytope theory.

2. The theoretical results are intuitive. The authors derive rates at which the success of model reconstruction changes in terms of the decision boundary complexity and the number of samples for linear models. Further they provide a bound on the model reconstruction error based on the observation that deep ReLU networks can be represented as Continuous Piece-wise Linear functions, whose decision boundaries form collections of polytopes.

3. The proposed approach provides state-of-the-art results on multiple benchmarks over the existing baseline for counterfactual-based model reconstruction.

**Weaknesses:**

1. Is there any dependency of the proposed method on the counterfactual generating method MCCF, or in other words, the quality/nature of the counterfactuals used?

2. A comparison of the proposed method with regular approaches (that treat counterfactuals and ordinary samples the same) like [Wang et al., 2022] using two sided counterfactuals could help solidify the contributions. I can understand that it might sound like an unfair comparison, but one should not necessarily expect the proposed CCA to outperform models using two-sided counterfactuals. This is just to get an understanding of how close can one get (using a model like CCA) to such two-sided counter-factual based approaches, by just using one-sided counterfactuals.

3. Comparing just with a single baseline seems a bit insufficient. Perhaps a discussion on how this work relates to model inversion attacks (such as [a,b]) would add to the completeness of the paper.

References:

[a] Zhao et al., "Exploiting Explanations for Model Inversion Attacks", ICCV 2021.

[b] Struppek et al. "Plug & Play Attacks: Towards Robust and Flexible Model Inversion Attacks", ICML 2022.

**Questions:**

Please refer to the Weaknesses section.

**Limitations:**

The limitations of this work have been adequately discussed.

---

> ### Author Rebuttal · Authors · 2024-08-07
>
> We are grateful for the detailed review and the important suggestions.
>
> **_Dependency on counterfactual generating method:_** The performance of the proposed method does not depend on the specific counterfactual generating method, except for the proximity of the generated counterfactuals to the decision boundary. CCA does not take any specifics of the generating method into account. However, the proximity to the decision boundary depends largely on the generating method. Therefore, we have considered several counterfactual generating methods in the experiments (Section D.2.6, Table 5 in manuscript). Furthermore, following the suggestions of Reviewer VLao, we have included two more counterfactual generating methods: **ROAR** [Upadhyay et al.] and **C-CHVAE** [Pawelczyk et al.] in the Rebuttal PDF (Table 7). In addition, we have included histograms of the predictions made by the target model on the generated counterfactuals and the query instances (see Rebuttal PDF Fig. 23).  They provide insights into the distribution of counterfactuals around the decision boundary. As evident from these, the CCA performance generalizes to a wide range of counterfactual generating methods as well as different counterfactual distributions w.r.t. the decision boundary.
>
> Moreover, we have discussed the effect of counterfactual sparsity on CCA performance in Section D.2.6 of the manuscript. It has been observed that the stricter the sparsity constraint gets, a significant amount of counterfactuals tend to lie further away from the decision boundary (Fig. 16). This in turn causes the gap between the baseline method and CCA to reduce (Fig. 17). We report some values in the table below for easy reference. An extreme effect of this nature can be observed in case of the ROAR generating method (Rebuttal PDF Table 7).
>
> Query size=100, fidelity over **test** data (psp=posthoc_sparsity_param – sparsity increases with increasing psp)
> |Model|psp=0.1|psp=0.2|
> |:---|:---:|:---:|
> |Base. M0|0.97|0.97|
> |CCA M0|0.98|0.96|
> |Base. M1|0.97|0.98|
> | CCA M1 |0.98|0.98|
>
> Query size=100, fidelity over **uniform** data
> |Model|psp=0.1|psp=0.2|
> |:---|:---:|:---:|
> |Base. M0|0.91|0.94|
> |CCA M0|0.92|0.90|
> |Base. M1|0.91|0.94|
> |CCA M1|0.94|0.95|
>
> **_Comparison with DualCFX_**: The related work [Wang et al.] is one of the few pioneering works studying the effects of counterfactuals on model extraction, which proposes the interesting idea of using counterfactuals of counterfactuals to mitigate the decision boundary shift. Based on the reviewer’s suggestion, we have now implemented DualCFX and included it in the Rebuttal PDF.
>
> The primary focus of our work is on the one-sided scenario where an institution might be giving counterfactuals to rejected applicants to help them get accepted only but not to the accepted ones. As the reviewer has correctly pointed-out, a fair comparison cannot be achieved between CCA and the strategy proposed in [Wang et al.] in the scenario where only one-sided counterfactuals are available since DualCFX requires counterfactuals from both sides.
>
> Therefore, in the two-sided scenario, we compare the performance of CCA with the DualCFX strategy proposed in [Wang et al.] under two settings:
> 1. only one sided counterfactuals are available for CCA (named CCA1);
> 2.  CCA has all the data that DualCFX has (named CCA2).
>
> We also include another baseline (following [Aïvodji et al.]) for the two-sided scenario where the models are trained only on query instances and counterfactuals, but not the counterfactuals of the counterfactuals. Results are presented in Table 6 of the Rebuttal PDF. Note that even for the same number of initial query instances, the total number of actual training instances change with the strategy being used (CCA1 < Baseline < DualCFX = CCA2 – e.g.: queries+CFs for the baseline but queries+CFs+CCFs for DualCFX).
>
> **_Related works in model inversion:_** We would like to thank the reviewer for suggesting two important related contributions. We will include a discussion on these works in a revised version of the manuscript.
>
> Model inversion is another form of extracting information about a black box model, under limited access to the model aspects. In contrast to model extraction where the goal is to replicate the model itself, in model inversion an adversary tries to extract the representative attributes of a certain class with respect to the target model. In this regard, [Zhao et al.] focusses on exploiting explanations for image classifiers such as saliency maps to improve model inversion attacks. [Struppek et al.] proposes various methods based on GANs to make model inversion attacks robust (for instance, to distributional shifts) in the domain of image classification.
>
>
> [Upadhyay et al.] Upadhyay, S., Joshi, S., & Lakkaraju, H. (2021). Towards robust and reliable algorithmic recourse. Advances in Neural Information Processing Systems, 34, 16926-16937.
>
> [Pawelczyk et al.] Pawelczyk, M., Broelemann, K., & Kasneci, G. (2020, April). Learning model-agnostic counterfactual explanations for tabular data. In Proceedings of the web conference 2020 (pp. 3126-3132).
>
> [Wang et al.] Wang, Y., Qian, H., & Miao, C. (2022, June). Dualcf: Efficient model extraction attack from counterfactual explanations. In Proceedings of the 2022 ACM Conference on Fairness, Accountability, and Transparency (pp. 1318-1329).
>
> [Aïvodji et al.] Aïvodji, U., Bolot, A., & Gambs, S. (2020). Model extraction from counterfactual explanations. arXiv preprint arXiv:2009.01884.
>
>  [Zhao et al.] Zhao, X., Zhang, W., Xiao, X., & Lim, B. (2021). Exploiting explanations for model inversion attacks. In Proceedings of the IEEE/CVF international conference on computer vision (pp. 682-692).
>
> [Struppek et al.] Struppek, L., Hintersdorf, D., Correira, A. D. A., Adler, A., & Kersting, K. (2022, June). Plug & Play Attacks: Towards Robust and Flexible Model Inversion Attacks. In International Conference on Machine Learning (pp. 20522-20545). PMLR.

---

> > ### Comment · Reviewer_Bnbn · 2024-08-12
> >
> > I thank the authors for their detailed response to my comments, which addresses all of my present concerns. I will carefully consider them to arrive at my final decision.

---

> > > ### Author Response · Authors · 2024-08-13
> > >
> > > Thank you very much for the insightful comments. We remain open to address any other comments and to provide any clarifications required.

---

### Official Review · Reviewer_VLao · 2024-07-12

**Soundness:** 3
**Presentation:** 3
**Contribution:** 3
**Rating:** 6
**Confidence:** 4

**Summary:**

This paper studies model reconstruction attacks by using the proximity of counterfactuals to the decision boundary. The authors aim to establish theoretical guarantees for such attacks. To this end, they characterize the number of queries required for the attacker to achieve a given error in model approximation using results from polytope theory (Theorem 2). The authors’ main result from relies on the decision boundary being convex. To relax the convexity assumption, the paper additionally studies the case of underlying relu networks to be attacked and provides probabilistic bound on the reconstruction rate. Finally, the authors propose a strategy for model extraction.

The paper offers strengths in terms of proposing new tools to analyze model extraction attacks. The main theoretical results are covered for two models classes that are commonly used in recourse literature (linear models, and NNs with relu activations) Overall, the paper provides a good starting point for future research in the area of model extraction attacks through counterfactual explanations but further improvements would be necessary to generealize the analysis to general commonly used models such a tree based classifiers.

**Strengths:**

- *New theoretical approach to study extraction attacks*: The paper introduces a fresh approach to studying model extraction attacks using counterfactual explanation algorithms, employing methodologies from polytope theory that I have not seen explored in this context before.
- *New method*: The authors propose a new model extraction method.
- *Clearly structured*: The paper is overall well written and clearly structured.

**Weaknesses:**

*Missing variety of recourse methods*: The paper would greatly benefit from a more comprehensive set of experiments that examine the viability of model reconstruction under various recourse settings. Specifically, incorporating experiments that generate counterfactuals with data manifold constraints [3,4] or consider robustness [1,2] would be highly valuable. Such experiments would clarify the conditions under which the proposed attacks are likely to succeed or fail, and could indicate potential defenses. For instance, if the attacks fail under certain counterfactual attack methods, this would highlight key characteristics necessary to ensure the safe use of recourse or counterfactual explanations. I strongly encourage the authors to include additional experiments addressing these aspects. Demonstrating the impact of these conditions would significantly strengthen the paper, and I would be happy to increase my score if such experiments are provided.

---------
**References**

[1] "Probabilistically Robust Recourse: Navigating the Trade-offs between Costs and Robustness in Algorithmic Recourse", ICLR, https://arxiv.org/abs/2203.06768

[2] "Towards robust and reliable algorithmic recourse.", NeurIPS, https://arxiv.org/abs/2102.13620

[3] "Learning model-agnostic counterfactual explanations for tabular data", WWW, https://arxiv.org/abs/1910.09398

[4] "Towards realistic individual recourse and actionable explanations in black-box decision making systems", arxiv, https://arxiv.org/abs/1907.09615

**Questions:**

Could the authors provide insights into their attempts to estimate the Lipschitz constant in practical applications? Considering the inherent difficulty in obtaining low Lipschitz constants for neural networks of reasonable size, the practicality of the Lipschitz result may be questionable. Clarifying this aspect would help in understanding the real-world applicability of their theoretical findings.

**Limitations:**

The authors did not mention any limitations of their work.

---

> ### Author Rebuttal · Authors · 2024-08-07
>
> We thank the reviewer for their review and greatly appreciate the positive opinion about our work.
>
> **_On the variety of recourse methods:_** As per the suggestions of the reviewer, we have now implemented **ROAR** [Upadhyay et al.] and **C-CHVAE** [Pawelczyk et al.]  methods and will include them in our paper.
>
> We considered the following counterfactual generating methods in the manuscript, including MCCF with L1 and L2 norms, DiCE actionable counterfactuals, 1-Nearest-Neighbor counterfactuals (counterfactuals from the data manifold) and DiCE counterfactuals with varying levels of sparsity (Section D.2.6 Table 5 and Fig. 16, 17). Now, we also include ROAR and C-CHVAE and present the consolidated results in **Rebuttal PDF** Table 7. We have also included histograms of the target prediction probabilities for counterfactuals generated by different methods.
>
> We would like to highlight the fact that our strategy does not take into account any specifics of the generating method. Instead, what affects the performance is the distribution of the counterfactuals around the decision boundary. Histograms in the Rebuttal PDF Fig. 23 provide some insights on how the counterfactuals generated using different methods are distributed. Firstly, observe that our strategy CCA does not require closest counterfactuals but is able to reconstruct models quite faithfully for many of the other counterfactual generating methods even if they are not the closest.
>
> Additionally, we also observe that the robust counterfactuals generated using the ROAR method have relatively higher prediction probabilities from the target model. As we may observe from these histograms, when the counterfactual distribution is concentrated around higher prediction probabilities (e.g.: in case of ROAR in the Rebuttal PDF or sparse DiCE with posthoc_sparsity_param=0.2 in the manuscript in section D.2.6. Fig. 16, 17), the advantage of CCA over the baseline diminishes.
>
> We will include a detailed discussion on these observations in the paper.
>
> **_On Lipschitz constants:_** In the experiments discussed in Section D.2.4 of the manuscript, we approximate the Lipschitz constant with the product of spectral norms of the weight matrices (following [Gouk et al., 2021]). A recent ICLR paper [Khromov and Singh] studies the tightness of this upper-bound as well as a lower-bound computed using the gradients around a subset of input instances. They point out that in practice, the actual Lipschitz constant lies more closer to the lower bound. They report the lower bound for a fully connected MLP with a layer width of 256 to be less than 40 [Khromov and Singh, Fig. 1].
>
> It is also noteworthy that Theorem 3.10 and Corollary 3.11 require the Lipschitz behavior only for a local region of any given input instance, provided that the counterfactuals are well-spread across the decision boundary. This local Lipschitz behavior is a much looser constraint than the global Lipschitzness.
>
> In essence, Lipschitzness attempts to capture the well-behavedness of the target model, in that it does not change very erratically. There may be scenarios where a model is reasonably well-behaved except in few regions which is captured by another body of work called probabilistic Lipschitzness. Such a probabilistic Lipschitz approach [Khan et al.] in conjunction with our result might generalize to a broader class of machine learning models, which can be a path for future exploration.
>
>
> [Upadhyay et al.] Upadhyay, S., Joshi, S., & Lakkaraju, H. (2021). Towards robust and reliable algorithmic recourse. Advances in Neural Information Processing Systems, 34, 16926-16937.
>
> [Pawelczyk et al.] Pawelczyk, M., Broelemann, K., & Kasneci, G. (2020, April). Learning model-agnostic counterfactual explanations for tabular data. In Proceedings of the web conference 2020 (pp. 3126-3132).
>
>  [Gouk et al., 2021] Gouk, H., Frank, E., Pfahringer, B., & Cree, M. J. (2021). Regularisation of neural networks by enforcing lipschitz continuity. Machine Learning, 110, 393-416.
>
> [Khromov and Singh] Khromov, G., & Singh, S. P. (2024). Some Fundamental Aspects about Lipschitz Continuity of Neural Networks. In The Twelfth International Conference on Learning Representations.
>
> [Khan et al.] Khan, Z. Q., Hill, D., Masoomi, A., Bone, J. T., & Dy, J. (2024, April). Analyzing Explainer Robustness via Probabilistic Lipschitzness of Prediction Functions. In International Conference on Artificial Intelligence and Statistics (pp. 1378-1386). PMLR.

---

> ### Comment · Reviewer_VLao · 2024-08-12
> **Response to Rebuttal**
>
> I very much appreciate the additional results provided by the authors. As the response has addressed all my concerns, I am increasing my score. I would expect the additional results to be discussed in the final version of the paper as well.

---

> > ### Author Response · Authors · 2024-08-13
> >
> > Thank you very much for updating the score. We will definitely include the additional results and a discussion in the updated version of the manuscript. We greatly appreciate the insightful review.

---

### Author Rebuttal · Authors · 2024-08-07

We thank all the reviewers for their insightful comments and suggestions. We are glad that our work has been recognized as an “in-depth analysis of a novel model reconstruction strategy” by Reviewer z1A7 and as a “good starting point for future works” by Reviewer VLao.

Model reconstruction using counterfactuals has received limited attention when compared to the vast literature of model extraction strategies using other cues. Within the scope of model extraction using counterfacuals, the existing works focus on scenarios where the counterfactuals are available from both sides of the decision boundary. Moving beyond this requirement, our work proposes a novel extraction strategy that utilizes counterfactuals lying only in the accepted region. We arrive at this strategy by analyzing such attacks through the lens of polytope theory. Moreover, we provide a comprehensive set of experiments comparing existing strategies with ours.

In addition to the individual responses/clarifications, we have conducted the following additional experiments as per the reviewer suggestions:
1. Two additional counterfactual generating methods (**ROAR** [Upadhyay et al.] and **C-CHVAE** [Pawelczyk et al.]) as suggested by Reviewer VLao
2. Comparing CCA with the **DualCFX** method proposed in [Wang et al.] as suggested by Reviewer Bnbn

Results of these experiments are included in the attached **Rebuttal PDF**.



[Upadhyay et al.] Upadhyay, S., Joshi, S., & Lakkaraju, H. (2021). Towards robust and reliable algorithmic recourse. Advances in Neural Information Processing Systems, 34, 16926-16937.

[Pawelczyk et al.] Pawelczyk, M., Broelemann, K., & Kasneci, G. (2020, April). Learning model-agnostic counterfactual explanations for tabular data. In Proceedings of the web conference 2020 (pp. 3126-3132).

[Wang et al.] Wang, Y., Qian, H., & Miao, C. (2022, June). Dualcf: Efficient model extraction attack from counterfactual explanations. In Proceedings of the 2022 ACM Conference on Fairness, Accountability, and Transparency (pp. 1318-1329).

---

### Decision · Program_Chairs · 2024-09-25

**Decision:**

Accept (poster)

**Comment:**

This paper Counterfactual Clamping that approximates the boundary of a convex classification region by counterfactual queries. The paper contains the performance guarantees for the fidelity measured between the true boundary and the approximated boundary. All four reviewers appreciate the new perspective of polytope theory for the (counterfactual) explanation problem. I thus recommend to accept this paper.